# Angiotensin-converting enzyme inhibitor promotes angiogenesis through Sp1/Sp3-mediated inhibition of notch signaling in male mice

Hanlin Lu[1], Peidong Yuan[1], Xiaoping Ma[1,2], Xiuxin Jiang[3], Shaozhuang Liu[3], Chang Ma[1], Sjaak Philipsen [4], Qunye Zhang [1], Jianmin Yang[1], Feng Xu[5], Cheng Zhang [1], Yun Zhang [1] & Wencheng Zhang [1] ✉

Angiogenesis is a critical pathophysiological process involved in organ growth and various diseases. Transcription factors Sp1/Sp3 are necessary for fetal development and tumor growth. Sp1/Sp3 proteins were downregulated in the capillaries of the gastrocnemius in patients with critical limb ischemia samples. Endothelial-specific Sp1/Sp3 knockout reduces angiogenesis in retinal, pathological, and tumor models and induced activation of the Notch1 pathway. Further, the inactivation of VEGFR2 signaling by Notch1 contributes to the delayed angiogenesis phenotype. Mechanistically, endothelial Sp1 binds to the promoter of Notch1 and inhibits its transcription, which is enhanced by Sp3. The proangiogenic effect of ACEI is abolished in Sp1/Sp3-deletion male mice. We identify USP7 as an ACEI-activated deubiquitinating enzyme that translocated into the nucleus binding to Sp1/Sp3, which are deacetylated by HDAC1. Our findings demonstrate a central role for endothelial USP7-Sp1/Sp3-Notch1 signaling in pathophysiological angiogenesis in response to ACEI treatment.

The vascular system is a dynamic network of blood vessels that constantly provides the growing organs in an embryo and the healing wounds with gases, liquids, nutrients, signaling molecules, and cells throughout the vertebrate body[1–3]. Endothelial cells (ECs) that line the inner surface of the blood vessels undergo a series of coordinated events, including cell activation, proliferation, migration, differentiation, branching, anastomosis, lumen formation, and remodeling into arteries and veins from preexisting ones, which is termed angiogenesis[2,4,5]. Insufficient angiogenesis contributes to ischemic disorders such as ischemic heart disease or critical limb ischemia; however, excessive and imbalanced angiogenesis leads to malignant, inflammatory, and retinal diseases[1,5]. Understanding the underlying mechanism is pivotal for developing effective therapeutics aimed at vascular disorders.

Transcription factor Sp1 and Sp3 play crucial roles in eukaryotic transcriptional machinery by binding to the GC-rich promoters of

[1]The Key Laboratory of Cardiovascular Remodeling and Function Research, Chinese Ministry of Education, Chinese National Health Commission and Chinese Academy of Medical Sciences, The State and Shandong Province Joint Key Laboratory of Translational Cardiovascular Medicine, Department of Cardiology, Qilu Hospital of Shandong University, Cheeloo College of Medicine, Jinan, China. [2]Department of Obstetrics and Gynecology, LiaoCheng People's Hospital, LiaoCheng, China. [3]Department of Bariatric and Metabolic Surgery, General Surgery, Qilu Hospital of Shandong University, Cheeloo College of Medicine, Jinan, China. [4]Department of Cell Biology, Erasmus MC, Rotterdam, The Netherlands. [5]Department of Emergency Medicine, Chest Pain Center, Shandong Provincial Clinical Research Center for Emergency and Critical Care Medicine, Qilu Hospital of Shandong University, Jinan, China. ✉e-mail: zhangwencheng@sdu.edu.cn

target genes[6-8]. Sp1-knockout mice did not survive beyond E10.5 and displayed a broad range of abnormalities[9]. Sp3-knockout mice showed growth retardation, which resulted in death prenatally or at birth due to complications, including cardiac malformations[10,11]. Recently, Sp1 and Sp3 seem to be closely related to angiogenesis, especially the vascular endothelial growth factors[12-15]. However, the role of endothelial Sp1/Sp3 in angiogenesis in vivo has not been well explored.

Among the different players of the angiogenic process, Notch families have emerged as a critical node, which is robustly expressed in the vascular system, consisting of four receptors (Notch1–4), three delta-like ligands (DLL1, 3, and 4), and two jagged-like ligands[16-18]. Although the role of Notch signaling in angiogenesis is clear, the regulation of Notch family members is not fully understood. Sp1 downregulated the expression of Notch signaling via lncRNA ZFAS1[19], and Notch ligand Ser is a target of Sp1[20]. Given the close link between Sp1 and Notch pathway in angiogenesis, we hypothesized that Sp1/Sp3 may regulate angiogenesis by modulating the transcriptional machinery essential for the Notch family.

Angiotensin-converting enzyme (ACE) is a dipeptidyl carboxypeptidase that catalyzes the conversion of inactive angiotensin I to angiotensin II and inactivates bradykinin (BK)[21,22]. ACE inhibitor (ACEI) is widely used to treat cardiovascular diseases and the efficacy of ACEI therapy for preventing hypertension has been well recognized[23]. Increasing evidence reveals that ACEI promotes angiogenesis in vivo[24-26]. Angiotensin (Ang) II activates Notch signaling via Ang II type 1 receptor (AT1)[27,28]. Renin-angiotensin system (RAS) seems to be closely related to Notch signaling[29]. Additionally, previous work has reported that ACEI improved VEGF levels in experimental diabetes[30] or in a mouse model of retinopathy[31]. However, there are findings suggesting that ACEI has the potential to inhibit tumor growth due to suppression of VEGF-induced angiogenesis[32,33], demonstrating that the function of ACEI in this setting remains unclear and requires further investigation. Therefore, given the recent findings, we sought to investigate whether ACEI regulates angiogenesis via Notch and VEGF signaling.

In this study, we demonstrate that ACEI promotes angiogenesis via upregulating Sp1/Sp3. A causal effect of Sp1/Sp3 on Notch1 was demonstrated in the endothelial cells, that is, Sp1/Sp3 downregulation led to the elevation of the Notch pathway.

## Results

### Endothelial Sp1/Sp3 is required for angiogenesis in vivo

To explore the role of Sp1/Sp3 in angiogenesis, we first assessed the expression of Sp1/Sp3 in human samples. Sp1/Sp3, which colocalized with the nucleus (DAPI) and CD31-labeled vascular endothelium, were significantly reduced around both the myofibers as well as the blood vessels in the gastrocnemius samples obtained from patients with CLI compared to that in healthy controls (Fig. 1A–C).

To investigate the role of endothelial Sp1/Sp3 in postnatal angiogenesis in mice, *VE-CAD-CreER^{T2+}/Sp1^{fl/fl}/Sp3^{fl/fl}* (dKO) mice were generated. We used a retinal angiogenesis model to examine angiogenic sprouting in the mouse retinas of P5 pups. Compared to the littermate *VE-CAD-CreER^{T2−}/Sp1^{fl/fl}/Sp3^{fl/fl}* (CTR) mice, dKO mice displayed a significant reduction in retinal vessel area, vasculature length, branching points, and the number of tip cells, tip sprouts, and filopodia in the sprouting region (Fig. 1D).

To demonstrate a more widespread role for endothelial Sp1/Sp3 in the reparative angiogenesis response to injury in vivo, we used a mouse hindlimb ischemia (HLI) model. After unilateral ligation and excision of the femoral artery, dKO mice exhibited a significantly impaired perfusion recovery (Fig. 1E) and angiogenesis (CD31^+ capillaries) compared to that in CTR mice (Fig. 1F). In addition, a cutaneous wound-healing model was used to estimate the host defense response to injury. Wound closure on the dorsal skin was almost complete in CTR mice on day 7. In sharp contrast,

this reparative process was significantly delayed in the dKO mice (Fig. 1G).

Tumor growth also depends on the outgrowth of blood vessels to form new vasculature, and is acknowledged as a model to explore angiogenesis. Consequently, we determined whether endothelial Sp1/Sp3 deletion alters tumor angiogenesis and tumor growth. Lewis lung carcinoma (LLC) cells were injected subcutaneously into the flanks of either CTR or dKO mice. After 16 days, dKO mice with LLC cells exhibited decreased tumor growth and tumor weight as compared to that in CTR mice (Fig. S1A–C). Meanwhile, decreased CD31-labeled blood vessels and increased HIF-1α-labeled hypoxic area were found in LLC tumors implanted into dKO mice (Fig. S1D, E). Similar results were observed in the subcutaneous B16 tumors (Fig. S2A–E) and subcutaneous MC38 tumors (Fig. S3A–E). Collectively, these data indicate that Sp1/Sp3 may play an essential role in angiogenesis.

### ACEI increases angiogenesis via Sp1/Sp3 in ECs

Given that data in Fig. 1 revealed downregulated Sp1/Sp3 in human and rodent led to attenuated angiogenesis conditions and ACEI promotes angiogenesis in vivo[25,26]. We investigated whether Sp1/Sp3 upregulation contributed to the ACEI-induced proangiogenic phenotype in postnatal angiogenesis. ACEI or vehicle was injected into CTR or dKO mice from P2 to P4. ACEI treatment substantially increased the retinal vessel area, vasculature length, branching points, and number of tip cells and tip sprouts in CTR mice but not in dKO mice (Fig. 2A). Therefore, we hypothesized that the proangiogenic effect of ACEI may be mediated by the Sp1/Sp3. An ex vivo model of aortic ring sprouting was used to verify the role of Sp1/Sp3 in the ACEI-mediated proangiogenic effects. Corresponding to the in vivo assay, the aortic rings from dKO mice did not respond to ACEI (Fig. 2B). Furthermore, we isolated mouse lung endothelial cells (MLECs) from CTR and dKO mice to demonstrate that Sp1 and Sp3 in ECs play a vital role in the proangiogenic effect of ACEI. In vitro capillary tube formation assay showed that Sp1/Sp3 deletion significantly inhibited the ACEI-induced increase in junction number and relative tube length (Fig. 2C).

To investigate the role of endothelial Sp1/Sp3 in adult pathological angiogenesis in the presence of ACEI, we used an HLI mouse model. The HLI model showed that the blood flow recovery and CD31^+ capillary density in the gastrocnemius were significantly increased by ACEI administration in the CTR mice. However, angiogenesis in the HLI model in the dKO mice showed no difference after ACEI administration (Fig. 3A, B). To further demonstrate the effect of ACEI on adult angiogenesis under pathological conditions, we employed a subcutaneous Matrigel plug angiogenesis assay. Strikingly, there was more red coloration in Matrigel plugs from ACEI-treated CTR mice, indicating the formation of a mass of new vessels. In contrast, Matrigel plugs from dKO mice, with or without ACEI administration, were relatively pale (Fig. 3C). The mRNA levels of EC markers CD31 and vascular endothelial-cadherin (VE-cadherin) and immunofluorescence staining of CD31^+ ECs also confirmed the effect of ACEI on angiogenesis (Fig. 3C). For tumor angiogenesis, there was almost no differences between mice treated with or without in tumor growth, tumor weight and vessel density (Fig. S4).

These results suggest that endothelial Sp1 and Sp3 play critical roles in ACEI-mediated physiological and pathological angiogenesis.

### ACEI stabilizes Sp1/Sp3 protein and promotes angiogenesis via USP7

Next, we asked how Sp1/Sp3 deletion abrogated the effect of ACEI in angiogenesis and whether ACEI directly modulates Sp1/Sp3 in endothelial cells. ACEI treatment increased the protein levels of Sp1/Sp3 in retinal ECs of CTR mice (Fig. S11A). This ACEI-mediated Sp1/Sp3 accumulation in ECs may be attributed to their posttranslational regulation. We focused on deubiquitinases and used siRNA to investigate the key deubiquitination enzymes. Among the dozens of

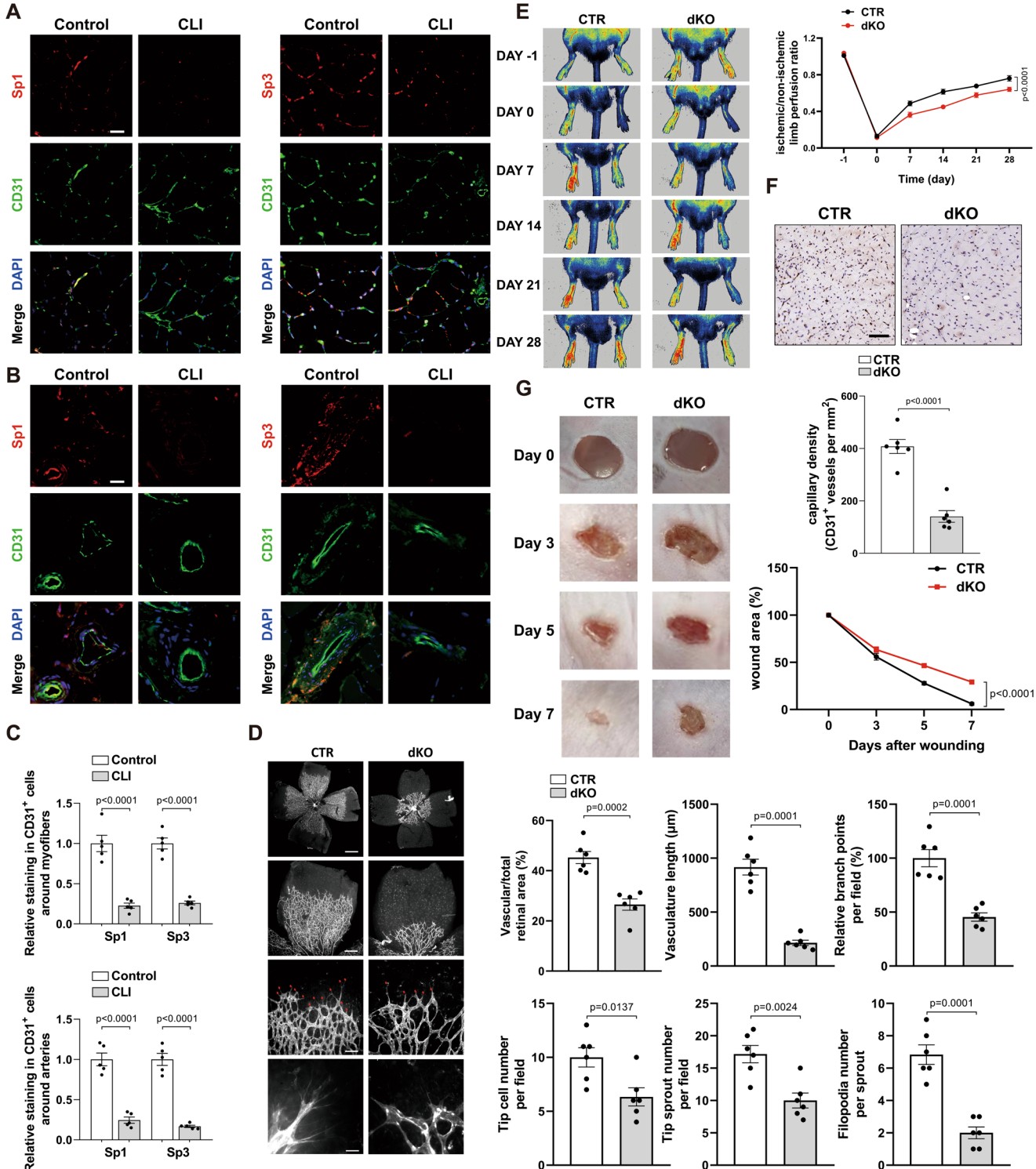

**Fig. 1 | Knockout of Sp1/Sp3 in ECs decreases angiogenesis in vivo.**
**A**, **B** Representative immunofluorescence staining of Sp1 and Sp3 in gastrocnemius muscle biopsies of patients with critical limb ischemia (CLI) compared to healthy subjects. CD31 is an endothelial cell marker. Scale bar: 20 μm. **C** Quantification of immunofluorescence staining of Sp1 and Sp3 in **A** and **B**. $n = 5$. **D** Retinal whole-mount staining of isolectin-B4 in P5 *VE-CAD-CreER^{T2-}/Sp1^{fl/fl}/Sp3^{fl/fl}* (CTR) and *VE-CAD-CreER^{T2+}/Sp1^{fl/fl}/Sp3^{fl/fl}* (dKO) mice. Red arrowheads show tip cell sprouting and filopodia. Right, quantification of vascular/total retinal area, vasculature length, branch points per field, tip cell number per field, tip sprout number per field, and filopodia number per sprout. Scale bars: 500 (row 1), 100 (row 2), 50 (row 3), and 10 μm (row 4). $n = 6$. **E** Representative laser Doppler images of the legs acquired on days −1 (before surgery), 0 (immediately after surgery), 7, 14, 21, and 28 after

surgery. Right, quantification of blood flow recovery after hindlimb ischemia as determined by the ratio of foot perfusion between ischemic (left) and non-ischemic (right) legs in CTR and dKO mice. $n = 6$. **F** Immunohistochemistry (IHC) analysis of CD31^+ staining (capillary density) in the ischemic gastrocnemius muscle. Bottom, quantification of CD31^+ vessels per mm^2 in CTR and dKO mice. $n = 6$. Scale bar: 50 μm. **G** Excisional cutaneous wounds were created using a 5 mm biopsy punch on the dorsal skin of CTR and dKO mice. $n = 6$. Bottom right, quantification of the wound area at days 0, 3, 5, and 7. Two-tailed Student's unpaired t-test was used for analysis in **C**, **D** and **F**. Two-way ANOVA followed by Bonferroni multiple-comparison analysis was used for **E** and **G**. Data are presented as mean ± SEM. Source data are provided as a Source Data file.

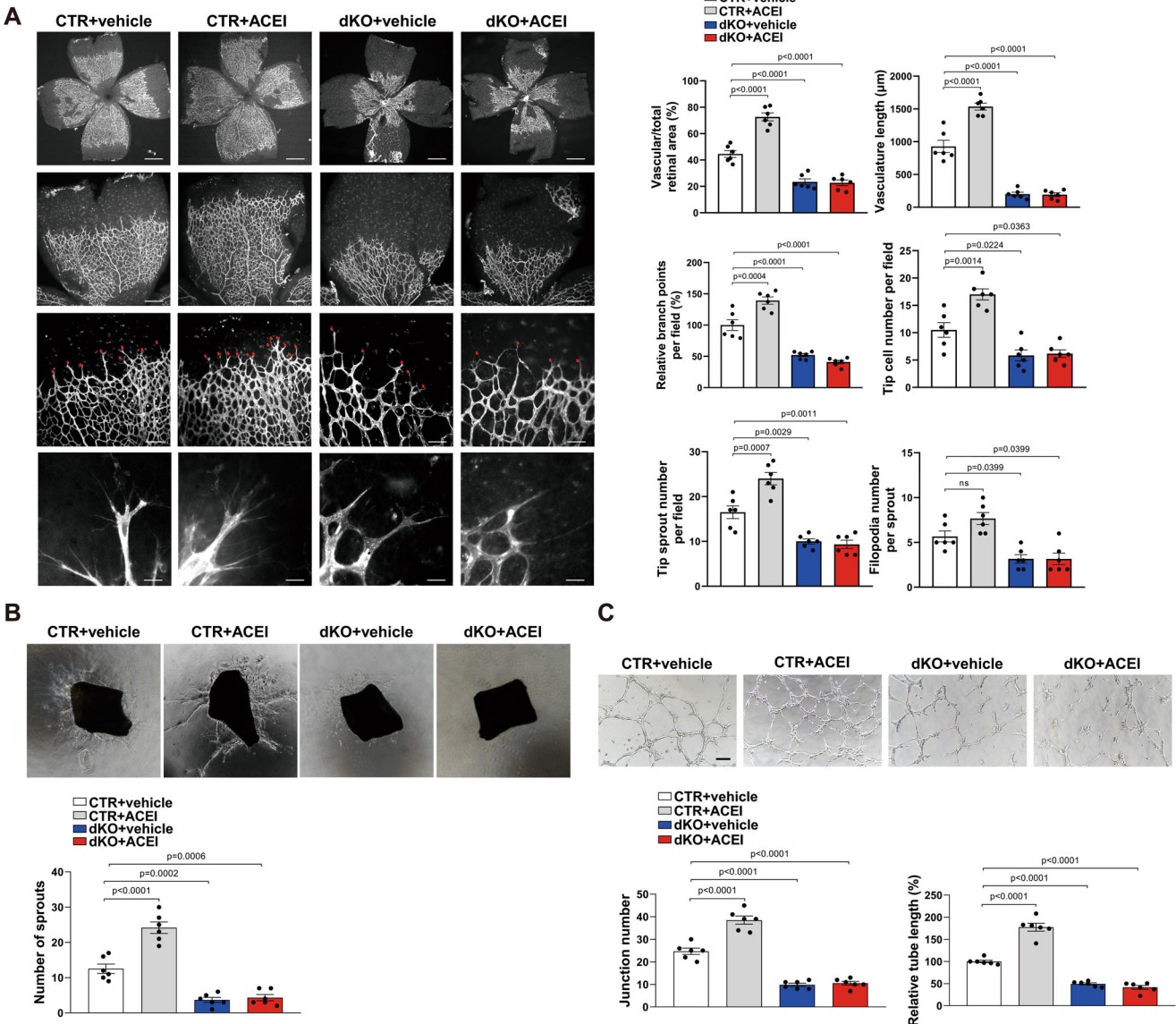

**Fig. 2 | ACEI promotes angiogenesis via Sp1/Sp3. A** Retinal whole-mount staining of isolectin-B4 in P5 CTR and dKO mice treated with the vehicle or angiotensin-converting enzyme inhibitor (ACEI). $n = 6$. Red arrowheads show tip cell sprouting and filopodia. Right, quantification of vascular/total retinal area, vasculature length, branch points per field, tip cell number per field, tip sprout number per field, and filopodia number per sprout. Scale bars: 500 (row 1), 100 (row 2), 50 (row 3), and 10 µm (row 4). $n = 6$. **B** Aortic ring assays showing the number of aortic sprouts from CTR and dKO mice treated with vehicle or ACEI embedded in Matrigel stimulated with vascular endothelial growth factor (VEGF, 20 ng/mL) for 6 days. Bottom, quantification of sprout number. $n = 6$. **C** Capillary network formation in mouse lung endothelial cells (MLECs) isolated from CTR and dKO mice treated with or without ACEI on Matrigel (VEGF stimulation for 4 h). Bottom, quantification of junction number and relative tube length. $n = 6$. Scale bar: 50 µm. Two-way ANOVA followed by Bonferroni multiple comparison analysis was performed for **A**–**C**. Data are presented as mean ± SEM. Source data are provided as a Source Data file.

deubiquitinases, only USP7-deficiency by siRNA reduced the protein levels of both Sp1 and Sp3 in human umbilical vascular endothelial cells (HUVECs) pretreated with ACEI (Fig. S5). Among the five distinct families of DUBs, USP7 belongs to the largest ubiquitin-specific protease (USP) family[34,35]. Other members, such as USP10, which are largely present in the cytoplasm, USP7 could exert its functions in the nucleus via translocation[34,36]. Ubiquitin-proteasome inhibitor MG132 attenuated the USP7 siRNA-induced Sp1/Sp3 decrease in HUVECs (Fig. S6A, B). In addition, USP7 knockdown significantly impaired Sp1/Sp3 stabilization when treated with cycloheximide compared to that in the control siRNA group (Fig. 4A).

To determine the contribution of USP7 to the enhanced retinal sprout angiogenesis observed in ACEI-treated mice, we tested the effects of P22077 and P5091, two selective USP7 inhibitors, in retinal angiogenesis. P22077, P5091, or vehicle were injected into the ACEI-treated group or littermate control group from P2 to P4. Retinal vessel development analyzed at P5 revealed that P22077- or P5091-treated mice exhibited delayed expansion of the vascular plexus, as evidenced by a decrease in the vascular area, vasculature length, and branching points. Further analysis revealed that at the leading edge of the developing vasculature, the number of tip cells, tip sprouts, and filopodia were reduced compared with those of littermates (Fig. 4B). The inhibitory effect of inactivated USP7 was also confirmed by the decreased junction number and relative tube length of HUVECs on Matrigel caused by P22077 or P5091 (Fig. 4C). To define the role of USP7 in angiogenesis under pathological conditions, we performed a Matrigel plug assay. The Matrigel plugs harvested from P22077- or P5091-treated mice displayed minimal vascularization compared with the excessive blood vessel formation (opaque plugs) from ACEI-treated mice (Fig. 4D). Further quantification of the vascular response

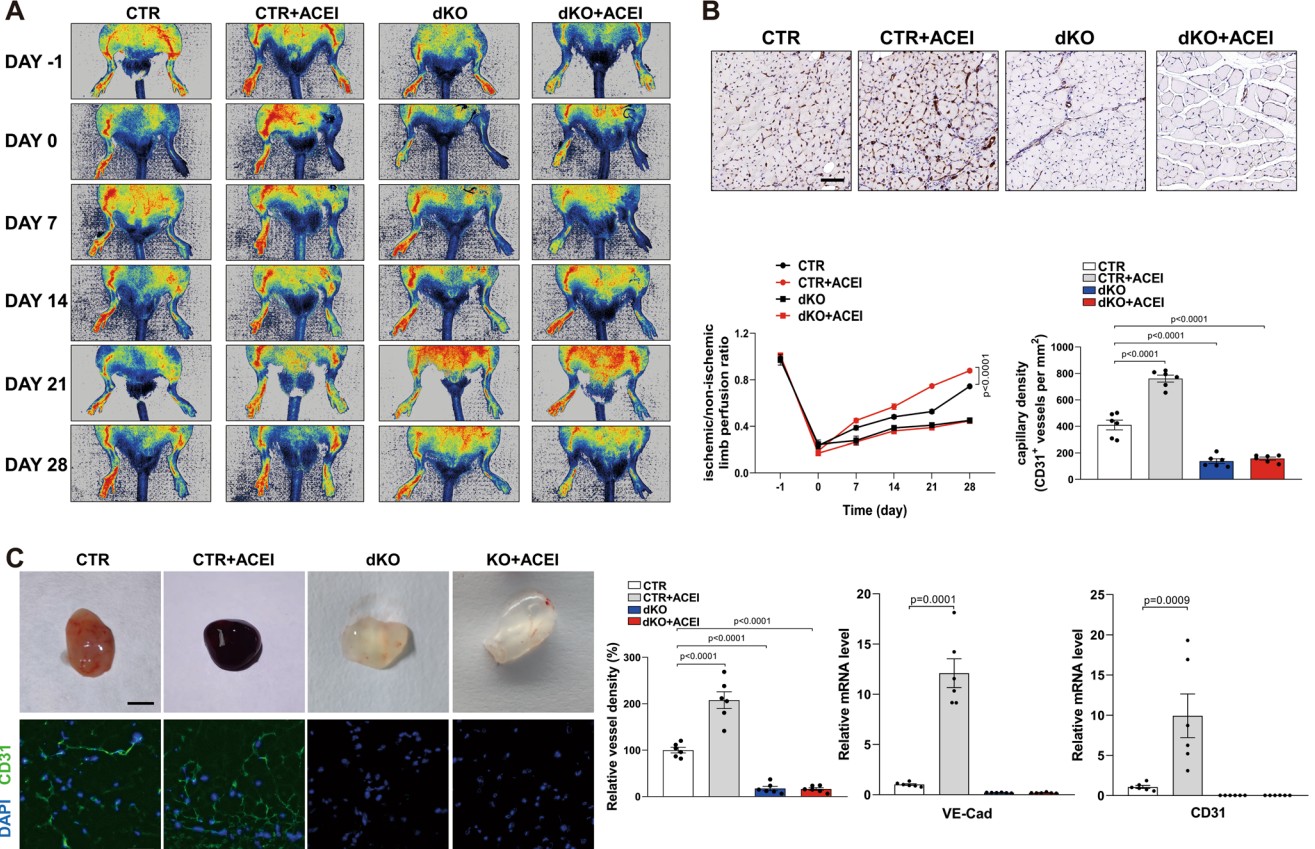

**Fig. 3 | ACEI promotes pathological angiogenesis via Sp1/Sp3. A** Representative laser Doppler images of the legs on day −1 (before surgery), 0 (immediately after surgery), 7, 14, 21, and 28. Right, quantification of blood flow recovery after hindlimb ischemia as determined by the ratio of foot perfusion of the ischemic (left) and non-ischemic (right) legs in CTR and dKO mice treated with or without ACEI. $n = 6$. **B** Assessment of capillary density by IHC analysis using CD31⁺ staining in the ischemic gastrocnemius muscle. Bottom, quantification of CD31⁺ vessels per mm² in CTR and dKO mice treated with or without ACEI. $n = 6$. Scale bar: 50 μm. **C** Images of the Matrigel plugs retrieved from CTR or dKO mice treated with or without ACEI and immunofluorescence staining of CD31⁺ vessels in the Matrigel plugs. Right, quantification of vessel density and mRNA expression levels of CD31 and VE-cadherin assessed using qPCR analysis. $n = 6$. Scale bars: 1 cm (row 1) and 20 μm (row 2). Two-way ANOVA followed by Bonferroni multiple comparison analysis was performed for **A–C**. Data are presented as mean ± SEM. Source data are provided as a Source Data file.

by evaluating the CD31⁺ stained vessels showed a decreased angiogenesis after P22077 or P5091 treatment. USP7-inhibition in mice also displayed a deleterious phenotype in HLI condition (Fig. S7). Collectively, these data provide compelling evidence that USP7-protected Sp1/Sp3 in endothelial cells contributes to the proangiogenic effect of ACEI.

## USP7 interacts with Sp1/Sp3 protein and decreases Sp1/Sp3 ubiquitination

USP7, a deubiquitinating enzyme, controls the protein level by direct deubiquitination[37–39]. Therefore, we hypothesized that USP7 regulates the stability of Sp1 and Sp3 by interacting with them leading to their deubiquitination. To test this hypothesis, we investigated the effect of USP7 on polyubiquitination of Sp1 and Sp3. In HEK293T cells, ectopic expression of USP7-WT, but not the catalytically inactive mutant USP7-CS (C233S), reduced the ubiquitination levels of Sp1 and Sp3 in the co-immunoprecipitation (Co-IP) assays (Fig. 5A). A similar trend was observed in HEK293T cells treated with USP7 inhibitors P22077 and P5091 (Fig. 5B). Next, we examined the specific roles of the catalytic site of USP7 in ACEI-induced angiogenesis response by overexpressing USP7-WT or USP7-CS in HUVECs. We observed a complete inhibition of ACEI-induced angiogenesis in USP7-CS compared to that in USP7-WT (Fig. S8A, B). These results indicated that the enzymatic effect of USP7 is essential for USP7-mediated deubiquitination of Sp1/Sp3.

To further verify which poly-ubiquitin chain on Sp1/Sp3 is removed by USP7, we transfected mutant-ub-K48R or mutant-ub-K63R, in which 48 or 63 lysine was mutated to arginine, into HEK293T cells. As shown in Fig. 5C, mutation of K48, but not K63, notably impaired the USP7 mediated Sp1/Sp3 deubiquitination, thereby confirming that USP7 stabilizes Sp1/Sp3 by removing the K48-linked poly-ubiquitin. To identify the USP7 domain involved in USP7 binding to Sp1/Sp3, we generated truncation mutants of Myc-USP7 that were co-transfected with Flag-Sp1 or Flag-Sp3 into HEK293T cells. To map the specific domain of USP7 responsible for USP7-Sp1 or USP7-Sp3, Co-IP assays were done, and an interaction between USP7 and Sp1/Sp3 dependent on the N-terminal TRAF-like domain (1–208 aas) of USP7 was found (Fig. 5D). In the quantitative ChIP assay, ACEI treatment enhanced the enrichment of Sp1 or Sp3 at the NOTCH1 promoter. In contrast, P22077 or P5091 treatment significantly impaired the increase observed in the ACEI-induced enrichment (Fig. 5E). Taken together, USP7 plays a critical role in ACEI-mediated upregulation in the transcriptional function of Sp1/Sp3 by increasing the accumulation of them.

## HDAC1 inhibits Sp1/Sp3 ubiquitination by deacetylating Sp1-lys703/Sp3-lys551

Acetylation has been identified as an evolutionarily conserved modification in proteins and plays important roles in protein stability[40,41]. Histone deacetylases (HDACs), recruited to gene promoters by several

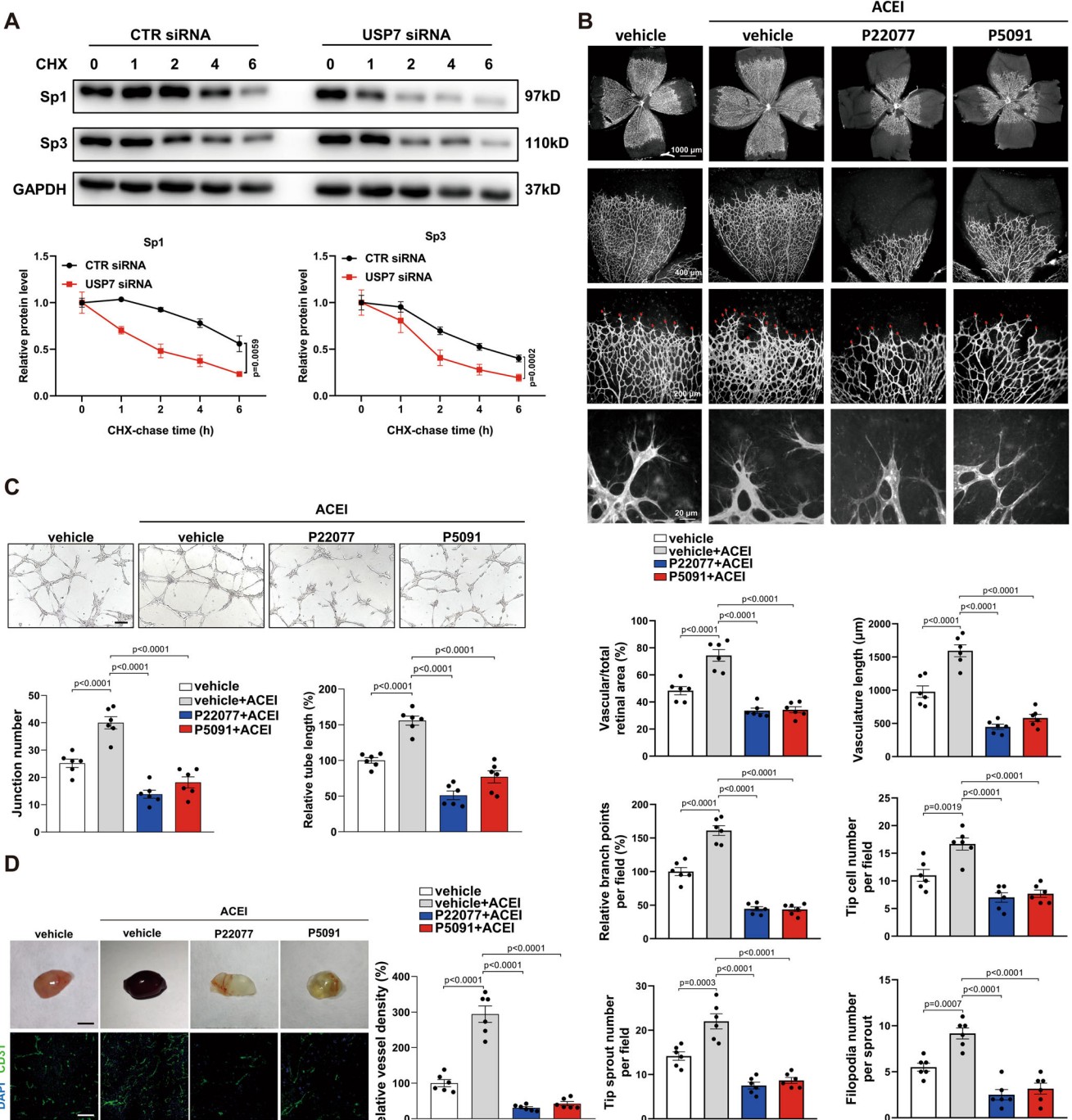

**Fig. 4 | ACEI promotes angiogenesis via USP7. A** Western blot analysis of Sp1 and Sp3 in human umbilical vascular endothelial cells (HUVECs) transfected with CTR siRNA or USP7 siRNA and treated with cycloheximide (CHX) for 0, 1, 2, 4, or 6 h. Bottom, quantification of western blot. $n = 6$. **B** Retinal whole-mount staining of isolectin-B4 in P5 pups treated with vehicle, vehicle + ACEI, P22077 + ACEI, or P5091 + ACEI. $n = 6$. Red arrowheads show tip cell sprouting and filopodia. Bottom, quantification of vascular/total retinal area, vasculature length, branch points per field, tip cell number per field, tip sprout number per field, and filopodia number per sprout. Scale bars: 500 (row 1), 100 (row 2), 50 (row 3), and 10 μm (row 4). **C** Capillary network formation of HUVECs with different treatments on Matrigel (VEGF stimulation for 4 h). Bottom, quantification of junction number and relative tube length. $n = 6$. Scale bar: 50 μm. **D** Images of the Matrigel plugs retrieved from mice treated with vehicle, vehicle + ACEI, P22077 + ACEI, or P5091 + ACEI, and immunofluorescent staining of CD31+ capillaries in the Matrigel plugs. Right, quantification of vessel density. $n = 6$. Scale bars: 1 cm (row 1) and 20 μm (row 2). Two-way ANOVA followed by Bonferroni multiple-comparison analysis for **A**. One-way ANOVA followed by Bonferroni multiple-comparison analysis was used for **B–D**. Data are presented as mean ± SEM. Source data are provided as a Source Data file.

mechanisms, including direct interaction with transcription factors like Sp1/Sp3[42,43]. To elucidate the mechanism of acetylation in Sp1/Sp3 stability regulation, HUVECs were treated with TSA or SAHA, inhibitors that block HDAC1. This treatment abrogated the increased interaction between USP7 and Sp1/Sp3 induced by ACEI (Fig. 5F).

Consistent with this result, when HDAC1 siRNA was transfected into HUVECs, the increased interaction between USP7 and Sp1 or Sp3 was impaired in comparison with that observed in CTR siRNA (Fig. 5G). In consideration of the promotional role of HDAC1 inhibition on the Sp1/Sp3 ubiquitination, we generated a Sp1 mutant (Flag-Sp1-K703A) and a

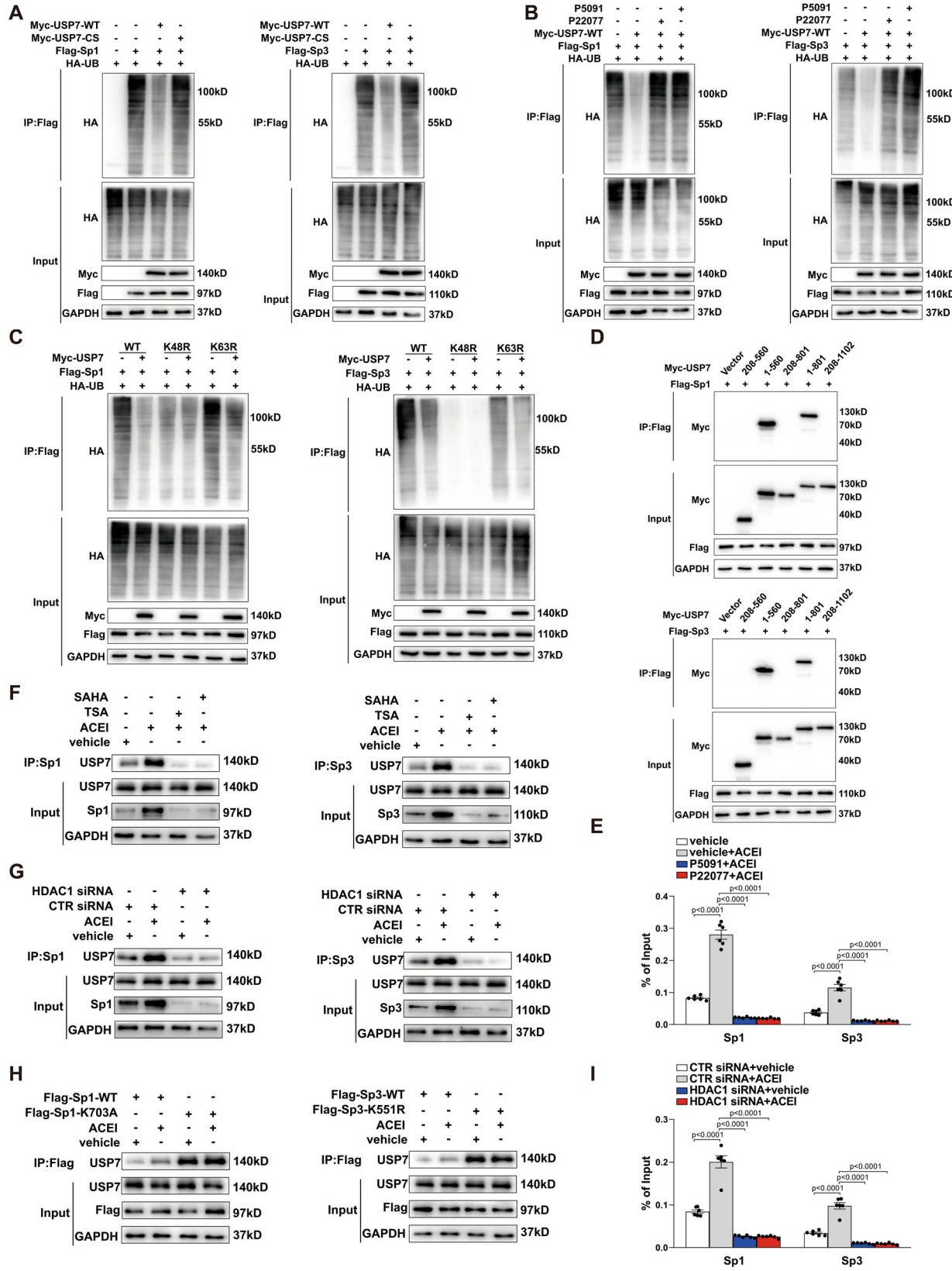

Sp3 mutant (Flag-Sp3-K551R) in which the acetylation sites Lys-703 and Lys-551 were mutated to alanine and arginine, respectively. Co-IP assays showed that Flag-Sp1-K703A or Flag-Sp3-K551R interacted more strongly with USP7 than wild-type Sp1 or Sp3, independent of ACEI treatment, supporting the notion that acetylation of Sp1/Sp3 prevented their binding with USP7 (Fig. 5H). In the quantitative ChIP assay,

HDAC1 knockdown resulted in decreased enrichment of Sp1 or Sp3 at the NOTCH1 promoter (Fig. 5I). These results clearly demonstrate that HDAC1-mediated deacetylation of Sp1/Sp3 in the presence of ACEI leads to their increased association with USP7. However, the underlying molecular mechanism of Sp1/Sp3 deacetylation in affecting its degradation via interaction with USP7 needs to be addressed.

**Fig. 5 | ACEI stabilizes Sp1/Sp3 protein via USP7-mediated deubiquitination.**
**A** Representative western blot of HA-ubiquitin after anti-Flag immunoprecipitation of HEK293T cells ectopically expressing Flag-Sp1 (left) or Flag-Sp3 (right) either alone or in combination with Myc-USP7-WT or Myc-USP7-CS (C233S).
**B** Representative western blot of HA-ubiquitin after anti-Flag immunoprecipitation of HEK293T cells ectopically expressing Flag-Sp1 (left) or Flag-Sp3 (right) either alone or in combination with Myc-USP7-WT treated with USP7 inhibitors P22077 or P5091. **C** Representative western blot of HA-ubiquitin after anti-Flag immunoprecipitation of HEK293T cells ectopically expressing Flag-Sp1 (left) or Flag-Sp3 (right), either alone or in combination with HA-ubiquitin-WT or mutant (K48R or K63R). **D** HEK293T cells ectopically expressing Flag-Sp1 (left) or Flag-Sp3 (right) were co-transfected with Myc-USP7 deletion mutants, as indicated. The interactions were analyzed using the Co- immunoprecipitation assay. **E** Chromatin immunoprecipitation (ChIP) assay showing binding of Sp1 or Sp3 to the NOTCH1 promoter in

HUVECs treated with ACEI either alone or in combination with P22077 or P5091. *n* = 6. **F** Representative western blot analysis for USP7 after anti-Sp1 (left) or anti-Sp3 (right) immunoprecipitation of HUVECs treated with ACEI either alone or in combination with the HDAC1 inhibitor SAHA or TSA. **G** Representative western blot analysis of USP7 after anti-Sp1 (left) or anti-Sp3 (right) immunoprecipitation of HUVECs transfected with CTR siRNA or HDAC1 siRNA, in combination with or without ACEI. **H** Representative western blot analysis of USP7 after anti-Flag immunoprecipitation of HUVECs ectopically expressing Flag-Sp1-WT (left, Flag-Sp3-WT) or Flag-Sp1-K703A (right, Flag-Sp3-K551R) treated with or without ACEI. **I** ChIP assay showing the binding of Sp1 or Sp3 to the NOTCH1 promoter in HUVECs transfected with CTR siRNA or HDAC1 siRNA, in combination with or without ACEI. *n* = 6. Two-way ANOVA followed by Bonferroni multiple-comparison analysis was employed for **E** and **I**. Data are presented as mean ± SEM. Source data are provided as a Source Data file.

## ACEI increases nuclear localization of USP7 independent of HDAC1 activity

We further hypothesized that the increased deubiquitination of Sp1/Sp3 in endothelial cells treated with ACEI was the result of nuclear localization of USP7. We treated HUVECs with ACEI and found that the level of USP7 in the nucleus rapidly increased with ACEI treatment at 30, 60, and 90 min (Fig. 6A). To further explore the nuclear binding of Sp1/Sp3 and USP7, we performed co-immunoprecipitation assay with Sp1 or Sp3 antibodies separately in the nuclear subcellular fractions. The increase in USP7 binding to Sp1/Sp3 was extended to 90 min without decline. The ubiquitination level of Sp1/Sp3 also decreased after ACEI treatment, with the USP7 translocation into the nucleus (Fig. 6A). Interestingly, despite increased acetylation of Sp1/Sp3 blocked their binding to USP7, the nuclear accumulation of USP7 induced by ACEI was not influenced by the knockdown of HDAC1 in HUVECs (Fig. 6A). Additionally, the subcellular distribution of USP7 was confirmed using immuno-fluorescence staining (Fig. 6B). Next, we examined the functional significance of Sp1 acetylation at Lys-703 and Sp3 acetylation at Lys-551 for ACEI-induced angiogenic responses in the ECs. Over-expression of Sp1-K703A, but not Sp1-WT inhibited HDAC1 siRNA-impaired HUVECs tube formation under ACEI treatment (Fig. 6C). The similar result was exhibited when Sp3-K551R overexpression in HUVECs (Fig. 6C).

It remains unclear how ACEI exerts effects on endothelial USP7. Phosphorylation of USP7 at serine 18 by CK2 is required for USP7 stability[36,44]. Additionally, CK2 inhibition leads to decreased angiogenesis[45–47]. Thus, we hypothesized that ACEI regulates USP7 via CK2-mediated phosphorylation. As shown in Fig. S9A, ACEI increased the phosphorylation level of USP7 in HUVECs, which was inhibited by CX-4945 (inhibitor against catalytic CK2α and CK2α' subunits). Further, CX-4945 attenuated the increased protein level of Sp1/Sp3 induced by ACEI (Fig. S9A) and blocked USP7 translocating into the nucleus (Fig. S9A, B).

Taken together, the above data indicate an accumulation of USP7 in the nucleus during the proangiogenic function of ACEI in ECs, where it binds to Sp1/Sp3 and protects Sp1/Sp3 from proteasomal degradation.

## Endothelial Sp1/Sp3 promotes angiogenesis via Notch1-VEGFR2 signaling

Notch signaling is one of the most important pathways involved in angiogenesis. During angiogenesis, DLL4 binding induces the pro-teolytic cleavage of Notch1 to release the Notch intracellular domain (NICD), which translocates into the nucleus to combine with its co-activators, thereby regulating the transcription of several genes related to angiogenesis, such as HEY1, HES1, and DLL4[17,48–50]. It suppresses angiogenesis in most vascular beds. In this regard, we examined the role of Sp1/Sp3 in the Notch pathway in the ECs. To address the potential link between Sp1/Sp3 and Notch

signaling in ECs, we extracted protein and mRNA from retinal ECs harvested from CTR and dKO mice at P5. Sp1/Sp3 deletion in ECs led to a significant increase in the protein levels of Notch1, NICD, and its target gene DLL4 (Fig. 7A). In addition, dKO retinal ECs displayed a significant upregulation of the mRNA expression levels of NOTCH1 and its main target downstream effectors HES1, HEY1 and DLL4 (Fig. 7B). In dKO retinas, Sp1/Sp3 deficiency-induced Notch1 upregulation was confirmed by immunofluorescence staining compared with that in CTR mice (Fig. 7C).

Based on the data obtained, we hypothesized that Notch signaling inhibition might rescue vascular defects in dKO retinas. CTR and dKO pups were treated with DAPT (N- [N-(3, 5-difluor-ophenacetyl)-l-alanyl]-s-phenylglycinet-butyl ester), a γ-secretase inhibitor, to test whether increased Notch1 activity was responsible for the observed phenotype in the dKO mice. In both CTR and dKO retinas, DAPT treatment significantly increased the vessel density and expansion of the vascular plexus in terms of retinal vessel area, vasculature length, branching points, and the number of tip cells, tip sprouts, and filopodia (Fig. 7E).

Furthermore, the rescue effect of DAPT in dKO mice was confirmed in the model of hindlimb ischemia. DAPT ameliorated the impairment of endothelial Sp1/Sp3 deletion in pathological angiogenesis caused by ischemia. From femoral artery ligation to day 28, perfusion recovery in DAPT-injected CTR and dKO mice was comparable (Fig. 7F). Additionally, IHC staining of CD31[+] capillaries in gastrocnemius harvested from dKO mice revealed a significant increase in the vascular density after DAPT treatment (Fig. 7G). VEGF signaling is a key driver of EC migration, proliferation, and capillary tube formation, primarily through VEGFR2[51]. Its impact on tip/stalk cell balance is mediated by Notch1 signaling[52]. We observed a decreased expression of p-VEGFR2 (Y1175) at the isolectin-B4 labeled leading vascular edge of the dKO retinas as assessed by immunofluorescence staining (Fig. 7D). In the LLC xenografted tumors of dKO mice, the expression of VEGFR2 was downregulated and Notch1 was upregulated in CD31-labled vessels compared with CTR mice (Fig. S10). Further, we found that ACEI almost completely inhibited Notch1 signaling in CTR retinal ECs, as evidenced by the protein and mRNA levels, but had no effect on Notch1 signaling in dKO mice (Fig. S11).

The role of Sp1/Sp3 in VEGF signaling was examined by treating HUVECs with mithramycin (Sp1 and Sp3 inhibitors). Treatment with mithramycin or NICD overexpression by Ad-NICD resulted in a remarkable decrease in VEGFR2 phosphorylation and its downstream signaling, p-PLCγ (Y783) and p-ERK1/2 (T202/Y204), induced by VEGF without affecting their protein expression in ECs (Fig. 7H). VEGF and angiopoietin-1 (Ang-1) are essential factors to promote angiogenesis through regulation of various signaling events in endothelial cells[53–55]. As shown in Fig. S12A, we isolated mouse lung endothelial cells (MLECs) from CTR and dKO mice. In vitro capillary tube formation assay showed that the combined action of VEGF and Ang1 promoted angiogenesis in dKO MLECs, suggesting that Sp1/Sp3 deletion just

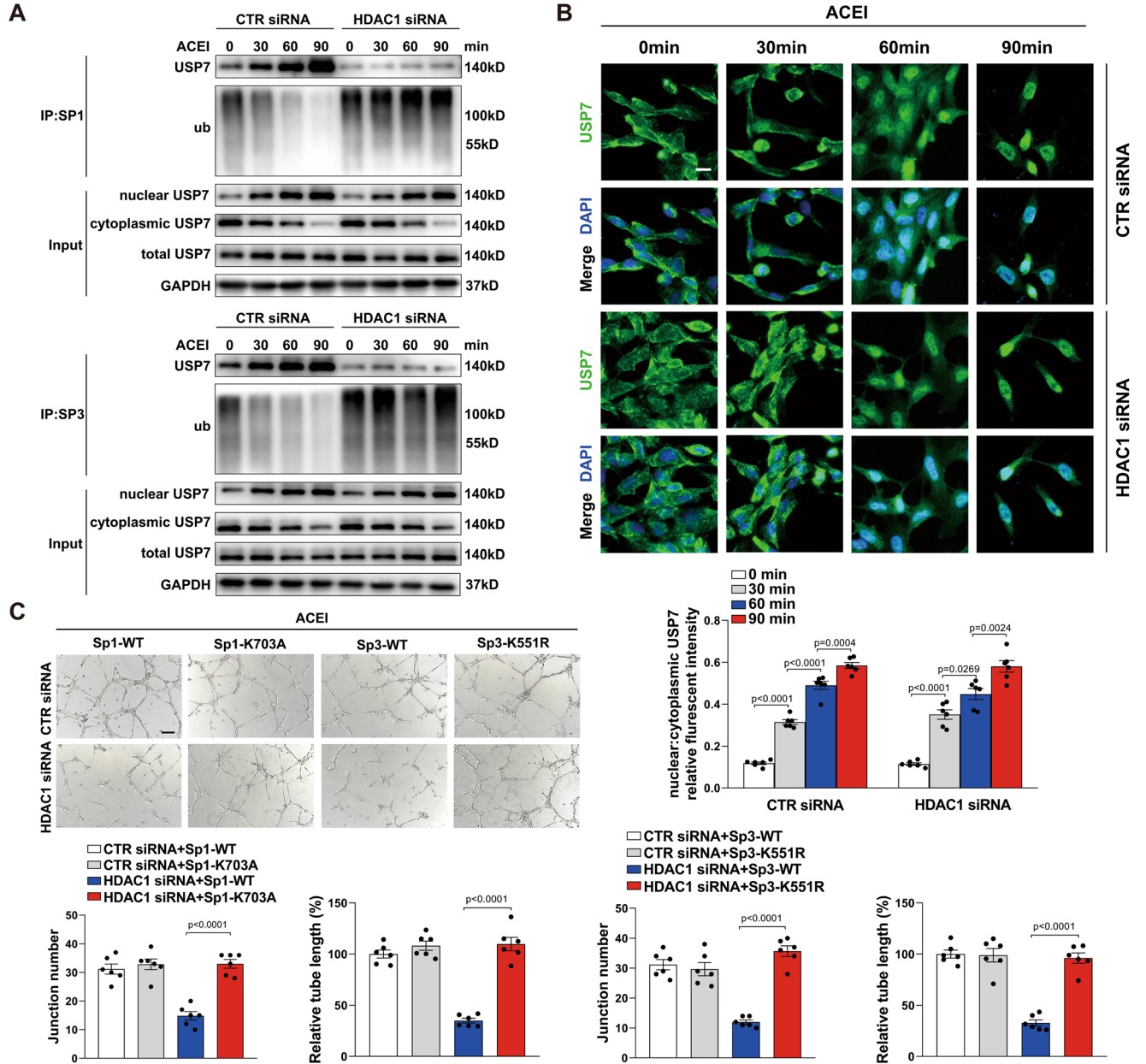

**Fig. 6 | ACEI increases nuclear localization of USP7. A** Representative western blot analysis of USP7 and ubiquitin after anti-Sp1 (top) or anti-Sp3 (bottom) immunoprecipitation of HUVECs treated with ACEI for different time, as indicated in combination with CTR siRNA or histone deacetylase 1 (HDAC1) siRNA. The protein levels of USP7 were measured by western blot analysis in nuclear, cytoplasmic, and whole-cell lysates. **B** Representative confocal microscopy images of immunofluorescence staining for USP7 and DAPI in HUVECs treated with ACEI for 0, 30, 60 and 90 min, in combination with CTR siRNA or HDAC1 siRNA as indicated. Scale bar: 20 µm. Bottom, quantification of the ratio of USP7 in the nuclear to cytoplasmic fractions. *n* = 6. **C** Capillary network formation in HUVECs on Matrigel treated with ACEI in combination with different genetic perturbations as indicated. Bottom, quantification of junction number and relative tube length. *n* = 6. Scale bar: 50 µm. One-way ANOVA followed by Bonferroni multiple-comparison analysis was employed for **B** and **C**. Data are presented as mean ± SEM. Source data are provided as a Source Data file.

blocked the VEGF signaling. Corresponding to the tube formation assay, the aortic rings from dKO mice also responded to VEGF and Ang1 ex vivo (Fig. S12B). Overexpression of NICD by Ad-NICD attenuated the activated VEGFR2 signaling caused by ACEI, suggesting that ACEI affects the activation of VEGFR2 via inhibiting Notch1 signaling rather than directly activating VEGFR2 (Fig. S13A). In HUVECs pretreated with ACEI, P22077 and P5091 inhibited the VEGF-induced phosphorylation of VEGFR2, PLCγ, and ERK1/2 considerably (Fig. S13B). Additionally, ACEI did not affect the expressions of angiogenic factors VEGFA and HIF1-α (Fig. S14).

Thus, our data suggested that Notch1-VEGFR2 signaling accounted for the mechanism of ACEI-Sp1/Sp3 regulating angiogenesis.

## Sp3 enhances Sp1-mediated transcriptional repression activity of NOTCH1 in ECs

Above data showed that Sp1 and Sp3 may work in the same transcription complex interactively. To further explore the distinct roles of Sp1 and Sp3, HUVECs were treated with Sp1 or Sp3 siRNA. The knockdown resulted in the upregulation of protein levels of Notch1, NICD, and DLL4 and the mRNA levels of NOTCH1, HES1, HEY1, and DLL4 (Fig. 8A, B). We then performed siRNA-mediated Sp1 knockdown in Sp3 overexpression (OE) ECs or Sp3 knockdown in Sp1 OE ECs. Sp1 knockdown in Sp3 OE ECs led to strong activation of Notch1 signaling (Fig. S15A, B) and increased mRNA levels (Fig. S15C). However, Sp3 siRNA-treated Sp1 OE ECs did not activate Notch1 signaling. These

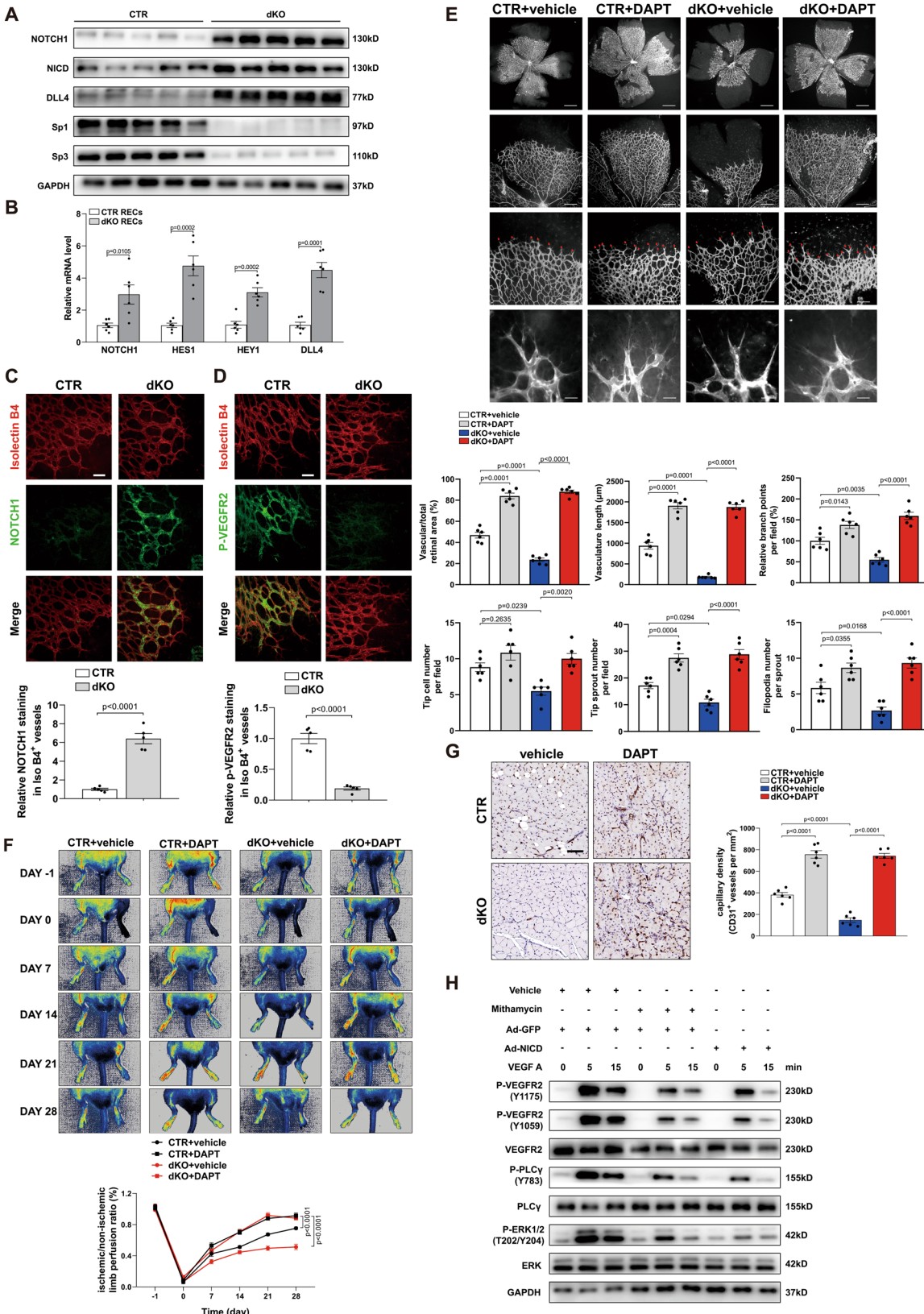

findings suggest that Sp3-mediated inactivation of Notch1 signaling depends on the presence of Sp1.

A simultaneous knockdown of expression using Sp1 and Sp3 siRNA induced further upregulation of Notch1 signaling, indicating a cooperative relationship between Sp1 and Sp3 (Fig. 8A, B). Sp1/Sp3 modulate transcription by binding to GC-box sequences in the

regulatory regions of its target genes. Several GC-box motifs are proximal to the NOTCH1 transcription start site (TSS). Chromatin immunoprecipitation (ChIP) was used to verify whether Sp1 and Sp3 interacted with the NOTCH1 promoter. Nuclear extracts from HUVECs were incubated with anti-Sp1 and anti-Sp3 antibodies or IgG (negative control). The ChIP primer for NOTCH1 was designed to amplify

**Fig. 7 | Endothelial Sp1/Sp3 negatively regulates Notch signaling to govern angiogenesis. A** Western blot analysis of Notch1, Notch intracellular domain (NICD), DLL4, Sp1, and Sp3 protein levels in isolated retinal endothelial cells from CTR and dKO mice. **B** qPCR analysis of NOTCH1, HES1, HEY1, and DLL4 mRNA levels in isolated retinal endothelial cells from CTR and dKO mice. $n = 6$. **C** Representative microscopy images and quantification of immunofluorescence staining for Notch1 in isolectin-B4$^+$ vessels of retinas from CTR and dKO mice. Scale bar: 30 μm. $n = 5$. **D** Representative microscopy images and quantification of immunofluorescence staining for phospho-VEGFR2 (Y1175) in isolectin-B4$^+$ vessels of retinas from CTR and dKO mice. Scale bar: 30 μm. $n = 5$. **E** Retinal whole-mount staining of isolectin-B4 in P5 CTR and dKO mice, with or without the DAPT (γ-secretase inhibitor). Red arrowheads show tip cell sprouting and filopodia. Bottom, quantification of vascular/total retinal area, vasculature length, branch points per field, tip cell number per field, tip sprout number per field, and filopodia number per sprout. Scale bars:

500 (row 1), 100 (row 2), 50 (row 3), and 10 μm (row 4). $n = 6$. **F** Representative laser Doppler images of legs on days −1 (before surgery), 0 (immediately after surgery), 7, 14, 21, and 28. Bottom, quantification of blood flow recovery after hindlimb ischemia as determined by the ratio of foot perfusion between ischemic (left) and non-ischemic (right) legs in CTR and dKO mice with or without DAPT. $n = 6$. **G** IHC analysis of CD31$^+$ staining (capillary density) in the ischemic gastrocnemius muscle. Right, quantification of CD31$^+$ vessels per mm$^2$ in CTR and dKO mice, with or without DAPT. Scale bar: 50 μm. $n = 6$. **H** Western blot analysis of phosphorylated VEGFR2 (Y1175, Y1059), VEGFR2, PLCγ (Y783), PLCγ, ERK1/2 (Thr202/Tyr204) and ERK1/2 in mithramycin or Ad-NICD treated HUVECs. Two-tailed Student's unpaired t-test was used for **B**–**D**. One-way ANOVA followed by Bonferroni multiple-comparison analysis was employed for **E** and **G**. Two-way ANOVA followed by Bonferroni multiple-comparison analysis was performed for **F**. Data are presented as mean ± SEM. Source data are provided as a Source Data file.

promoter regions containing putative binding sites for Sp1 and Sp3. We observed that both Sp1 and Sp3 could bind to the NOTCH1 promoter (Fig. 8C).

To further understand the transcriptional regulation of NOTCH1 by Sp1 and Sp3, we cloned different truncated promoters of NOTCH1 upstream of the luciferase reporter gene. NOTCH1 promoter activity was observed in the −68 to −35 segment, suggesting a role for this minimal Ad-Sp1 and Ad-Sp3-responsive element in regulating NOTCH1 transcription (Fig. 8D). Meanwhile, overexpression of Sp1 caused more severe inhibition of NOTCH1 transcription than Sp3, in line with the previous findings (Fig. 8D). To further verify the putative transcription factor binding sites for Sp1 and Sp3 in the NOTCH1 promoter, we prepared a substitution mutation of an individual site, which resulted in a significant decrease in transcriptional inhibitory activity compared to that of the wild-type promoter (Fig. 8E). Additionally, it was demonstrated that knockdown of HEY1/HES1 by siRNA attenuated Ad-NICD-induced repression of the VEGFR2 promoter (Fig. 8F). These results demonstrated that Sp1/Sp3 bind to the NOTCH1 promoter and inhibits its transcription.

To determine the independent roles of endothelial Sp1 and Sp3 in promoting angiogenesis, we analyzed retinal vessel development at P5 in *VE-CAD-CreER*$^{T2+}$/*Sp1*$^{fl/fl}$ (Sp1$^{ECKO}$) and *VE-CAD-CreER*$^{T2+}$/*Sp3*$^{fl/fl}$ (Sp3$^{ECKO}$) mice. In accordance with the in vitro data, retinas from Sp1$^{ECKO}$ mice and Sp3$^{ECKO}$ mice showed a mild delay in vasculature development phenotype compared with that of dKO mice. Furthermore, retinas from Sp1$^{ECKO}$ mice exhibited more severely impaired angiogenesis than Sp3$^{ECKO}$, indicating the prior role of Sp1 in angiogenesis to Sp3 in ECs (Fig. 8G). In addition, the CTR and dKO mice showed no difference in the vessel number of limb muscle before ligating the artery (Fig. S16A), the vascular permeability (Fig. S16B), and the basal inflammation level in serum (Fig. S16C).

Collectively, these findings suggest that Sp1-mediated transcriptional repression activity of NOTCH1 in ECs can be enhanced by Sp3.

## Discussion

Ischemic cardiovascular disease, typified by stroke, CLI, and MI, has become a major cause of global morbidity and mortality[56]. Angiogenesis is critical to maintaining vessel homeostasis under both physiological and pathological conditions, including development, wound healing, hypoxia, and ischemia, which is decisive for both host repair and therapeutic intervention. The transcription factors Sp1/Sp3 play important roles in various cardiovascular diseases and therefore have been a major research focus. In CLI patient samples, we found that the expression of Sp1 and Sp3 in vascular endothelium was dramatically decreased around both myofibers and artery vessels. Recently, Sp1 and Sp3 seem to be closely related to angiogenesis, especially the transcription of VEGF[12–15]. However, none of these studies specifically assessed the effect of Sp1/Sp3 deletion in endothelial cells. In this study of tamoxifen-inducible CreER$^{T2}$-mediated deletion of endothelial Sp1/Sp3 mice, the loss of Sp1/Sp3 delayed the angiogenesis in neonatal

retinas, reparative angiogenesis to HLI and skin wound, even the angiogenesis in the subcutaneous tumor. Moreover, we separately established Sp1$^{ECKO}$ and Sp3$^{ECKO}$ mice to explore the similarities and differences between these two relevant transcription factors. Therefore, Sp1/Sp3 might be promising angiogenic promoters for clinical use in impaired angiogenesis diseases.

The major finding of this study was that transcription factors Sp1/Sp3 are targets of ACEI in endothelium. Traditionally, ACEI is considered by inhibiting angiotensin II formation but also via BK accumulation to suppress the progress of cardiovascular disease. ACE inhibition increases the capillary density in stroke-prone spontaneously hypertensive rats and promotes angiogenesis in ischemic rabbit hindlimbs[24,25]. The proangiogenic effect of ACEI in vivo was abolished by blockage or deletion of the bradykinin B2 receptor[24,26]. However, the specific downstream signaling of ACEI in angiogenesis in the ECs is unknown. Herein, we found that administration of ACEI significantly improved angiogenesis and exhibited protective effects in CTR mice, but was unable to rescue the adverse phenotype caused by the deletion of endothelial Sp1/Sp3. Diminution of endothelial Sp1/Sp3 may therefore underpin the inability of ACEI to promote angiogenesis to maintain appropriate blood supply.

Protein ubiquitination is an important regulatory mechanism affecting protein stability, localization, and activity[57]. USP7 is a member of a deubiquitinating enzyme family that contains more than 90 genes[58]. Recently, USP7 was reported to be an important regulator of transcription factors[39,59,60]. Our studies indicated an accumulation of USP7 in the nucleus during the proangiogenic function of ACEI in ECs, where it binds to Sp1/Sp3 and protects Sp1/Sp3 from proteasomal degradation. Furthermore, USP7 inhibitor significantly abrogated the proangiogenic function of ACEI, indicating the adverse effect of potential drug aiming at USP7[61]. Phosphorylation of the Ser18 site by the kinase CK2 stabilizes USP7, thus contributing to the downstream stability of MDM2 in the nucleus[36]. In support of this, our results indicate that CK2 inhibitor treatment abrogates ACEI-induced USP7 from the cytoplasm to the nucleus. Additionally, ACEI decreased Sp1/Sp3 acetylation by activating HDAC1, which promoted the nuclear Sp1/Sp3 interaction with USP7. Interestingly, inhibition or knockdown of HDAC1 abrogated the interaction between Sp1/Sp3 and USP7, but showed no effect on USP7 transport from the cytoplasm induced by ACEI. Our observations uncovered a mechanism of ACEI and the regulation of Sp1/Sp3 in ECs.

Mechanistically, we establish a previously undefined role for endothelial Sp1/Sp3 in angiogenesis. This critical mechanism is dependent on transcriptional suppression of Notch1, leading to stimulation of the VEGF-VEGFR2 pathways. Angiogenesis requires the coordinated behaviour of endothelial cells, regulated by Notch and VEGF-VEGFR signaling. VEGF is a key regulator of physiological and pathological angiogenesis during various biological processes[62]. The Notch signaling pathway has been identified as a prominent negative modulator of angiogenesis counteracting VEGF signaling[63,64].

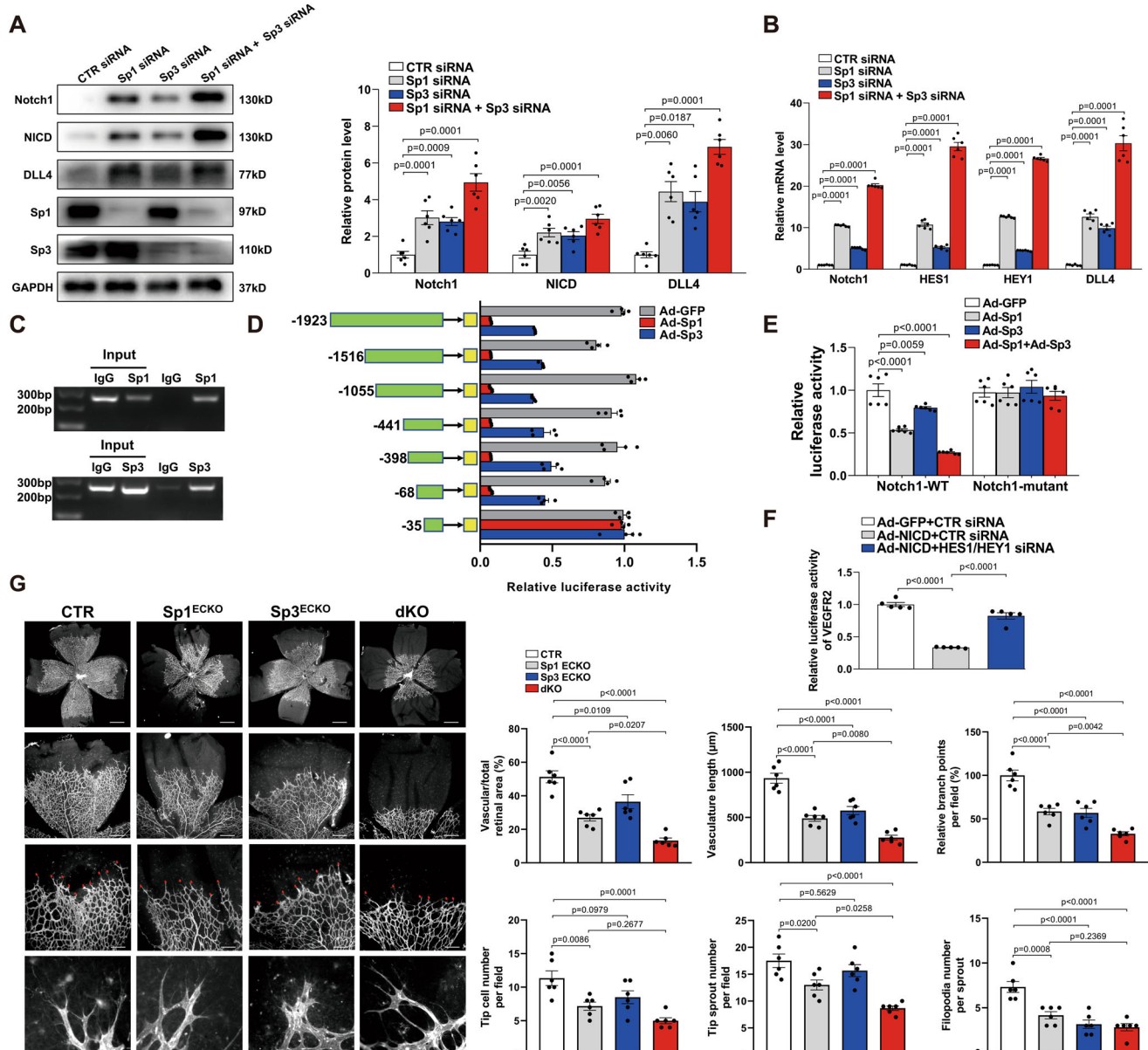

**Fig. 8 | Sp3 enhances Sp1-mediated transcriptional repression activity of NOTCH1 in ECs. A** Western blot analysis of Notch1, NICD, DLL4, Sp1, and Sp3 protein levels in HUVECs transfected with CTR, Sp1, Sp3, or Sp1 + Sp3 siRNA. *n* = 6. **B** qPCR analysis of NOTCH1, HES1, HEY1, and DLL4 mRNA levels in HUVECs transfected with siRNAs for CTR, Sp1, Sp3, or Sp1 + Sp3. *n* = 6. **C** Chromatin immunoprecipitation (ChIP) assay showing the binding of Sp1 or Sp3 to the NOTCH1 promoter in HUVECs. **D** Relative luciferase activity was shown by the indicated serial promoter deletions of NOTCH1 in HEK293T cells infected with Ad-GFP, Ad-Sp1, or Ad-Sp3. *n* = 6. **E** Relative luciferase activity of wild-type (NOTCH1-WT) and mutant constructs (NOTCH1-mutant) of the NOTCH1 promoter in HEK293T cells infected with Ad-GFP, Ad-Sp1, Ad-Sp3, or Ad-Sp1+Ad-Sp3. *n* = 6. **F** Relative luciferase activity of VEGFR2 promoter in HUVECs with different treatments. *n* = 5. **G** Retinal whole-mount staining of isolectin-B4 in P5 CTR, Sp1^ECKO, Sp3^ECKO, and dKO mice. *n* = 6. Red arrowheads show tip cell sprouting and filopodia. Right, quantification of vascular/total retinal area, vasculature length, branch points per field, tip cell number per field, tip sprout number per field, and filopodia number per sprout. Scale bars: 500 (row 1), 100 (row 2), 50 (row 3), and 10 μm (row 4). One-way ANOVA followed by Bonferroni multiple-comparison analysis was used for **A–G**. Data are presented as mean ± SEM. Source data are provided as a Source Data file.

Endothelial tip cells are induced and guided by an extracellular gradient of VEGF, which guides the growth of new vascular sprout by integrating attractive and repulsive environmental cues[65]. To prevent other ECs from turning into tip cells, a negative feedback regulation is imposed by Notch-mediated lateral inhibition, which down-regulates VEGF signaling in adjacent stalk cells to ensure the proper formation of a new vascular sprout connected to the existing vasculature[66]. It has been previously reported that Sp1/Sp3 are required for transactivation of VEGF promoter, dependent of proximal GC-rich boxes[12]. However, whether Sp1/Sp3 regulates Notch1 transcription has not been explored. In this study, loss of transcriptional suppression caused by

Sp1/Sp3-deficiency leads to overexpression of Notch1. In addition, Sp3-mediated inactivation of Notch1 signaling depending on the presence of Sp1 was proved by separate experiments. Collectively, our findings demonstrate that upregulation of Sp1/Sp3 under ACEI promotes angiogenesis through reciprocal cross-talks with the VEGF and Notch signaling pathways by augmenting VEGF signaling while dampening Notch activation.

In conclusion, our findings demonstrate that endothelial Sp1/Sp3 plays a vital role in angiogenesis by inhibiting NOTCH1 transcription and provides insights into endothelial Sp1/Sp3 as a potential target for therapy in various angiogenesis-related disorders, including retinal

diseases, CLI, and even tumors. Furthermore, the connection between the proangiogenic effect of ACEI and the Notch/VEGFR2 system is described by HDAC1-mediated deacetylation and USP7-mediated deubiquitination of Sp1/Sp3, which contributes to the investigation of the pharmacological effect of ACEI.

## Methods

### Ethical statement
Animal experiments were approved by the Animal Care Committee of Shandong University and were performed in compliance with the Animal Management Rules of the Chinese Ministry of Health. All animal experiments conformed to the guidelines of Directive 2010/63/EU of the European Parliament on the protection of animals used for scientific purposes. All ethical guidelines were adhered to whilst carrying out this study.

### Statistics
Results are expressed as the mean ± SEM, and n denotes the number of animals or independent experiments per group. Each experiment has performed a minimum of five times to ensure the reproducibility of the results. No samples or animals were excluded from the analysis. Statistical analyses were performed using GraphPad Prism version 8 (GraphPad Software, San Diego, CA, USA, www.graphpad.com) using Student's t-test and one-way or two-way ANOVA with Bonferroni post hoc tests. $P < 0.05$ was considered statistically significant.

### Antibodies and reagents
Antibodies against GAPDH (#5173, dilution: 1:1000), Notch1 (#3608, dilution: 1:1000), DLL4 (#96406, dilution: 1:1000), cleaved Notch1 (NICD, #4147, dilution: 1:1000), phospho-VEGF Receptor 2 (Tyr1175) (#2478, dilution: 1:1000), phospho-VEGF Receptor 2 (Tyr1059) (#3817, dilution: 1:1000), VEGF receptor 2 (#2479, dilution: 1:1000), phospho-p44/42 MAPK (ERK1/2) (Thr202/Tyr204) (#4370, dilution: 1:1000), p44/42 MAPK (Erk1/2) (#4695, dilution: 1:1000), phospho-PLCγ1 (Tyr783) (#14008, dilution: 1:1000), PLCγ1 (#5690, dilution: 1:1000), rabbit IgG (#7074), ubiquitin (#3936, dilution: 1:1000), FLAG (#14793, dilution: 1:1000), HA (#3724, dilution: 1:1000), Myc (#2276, dilution: 1:1000), HIF-1α (#14179, dilution: 1:1000), were from Cell Signaling Technology (Boston, MA, USA). USP7 (A300-033A-T, dilution: 1:1000) was from Bethyl. Sp1 antibody (07-645, dilution: 1:1000) was from Millipore. Sp3 (ab227856, dilution: 1:1000), VEGFA (ab46154, dilution: 1:1000) and CD31 (ab28364, dilution: 1:100) were from Abcam. Phosphoserine/threonine antibody (PP2551) was from ECMbiosciences. HRP-conjugated Affinipure Goat Anti-Mouse IgG (H + L) (SA00001-1, dilution: 1:5000) and HRP-conjugated Affinipure Goat Anti-Rabbit IgG (H + L) (SA00001-1, dilution: 1:5000) were from Proteintech. Perindopril (PHR1887), trichostatin A (TSA, T1952), SAHA (SML0061), DAPT (565770), tamoxifen (T5648), VEGF (V5765), were from Sigma-Aldrich (USA). Mithramycin A (MITA, HY-A0122), cycloheximide (CHX, HY-12320), CX-4945 (HY-50855), P22077 (HY-13865) and P5091 (HY15667) were from MedChemExpress (China). MG132 (S2619) was from Selleckchem (USA). Isolectin B4-594 (I21413) was from Invitrogen. Recombinant human angiopoietin-1 was from R&D System (R&D System, USA).

### Animal studies
Mice expressing the Cre recombinase under control of the VE-Cadherin promoter/ enhancer (*VE-CAD-CreER^{T2}*) were a gift from Prof. Yulong He in Soochow University[67]. Sp1-floxed (*Sp1^{fl/fl}*) and Sp3-floxed (*Sp3^{fl/fl}*) mice were obtained from Prof. Sjaak Philipsen (Erasmus University Medical Center, Rotterdam)[68]. To ablate Sp1 and Sp3 specifically in the endothelium (Sp1/Sp3^{ECKO}), *Sp1^{fl/fl}* and *Sp3^{fl/fl}* mice were crossbred with *VE-CAD-CreER^{T2}* mice to obtain *VE-CAD-CreER^{T2+}*/

*Sp1^{fl/fl}/Sp3^{fl/fl}* mice, which were intraperitoneally injected with tamoxifen (50 mg/kg) for 5 consecutive days. Littermate *VE-CAD-CreER^{T2}*/ *Sp1^{fl/fl}/Sp3^{fl/fl}* mice were treated with the same dose of tamoxifen as controls (CTR). To induce the deletion of endothelial Sp1 and Sp3 in pups, we administered tamoxifen (1 mg/mL, 50 μL) by intraperitoneal injection into each pup from P2 to P4.

For ACEI treatment, 8-week-old mice were injected intraperitoneally with an ACEI (perindopril, 3 mg/kg/d) for 28 days in hindlimb ischemia model or for 14 days in a Matrigel plug assay. In the postnatal retinal angiogenesis model, ACEI (perindopril, 1 mg/mL, 10 μL) was administered by intraperitoneal injection to each pup from P2 to P4.

For DAPT treatment, 8-week-old mice were injected intraperitoneally with DAPT (60 mg/kg/d) for 28 days in hindlimb ischemia model. In the postnatal retinal angiogenesis model, DAPT (20 mg/mL, 10 μL) was administered by intraperitoneal injection to each pup from P2 to P4.

For USP7 inhibitor treatment, 8-week-old mice were injected intraperitoneally with P22077 (15 mg/kg/d) or P5091 (15 mg/kg/d) for 28 days in hindlimb ischemia model or for 14 days in a Matrigel plug assay. In the postnatal retinal angiogenesis model, P22077 (5 mg/mL, 10 μL) or P5091 (5 mg/mL, 10 μL) was administered by intraperitoneal injection to each pup from P2 to P4.

Mice were housed at 25 °C, 12 h light/dark and were euthanized using an overdose of anesthesia with 1–1.5% isoflurane, followed by exsanguination and tissue removal. Animal experiments were approved by the Animal Care Committee of Shandong University and were performed in compliance with the Animal Management Rules of the Chinese Ministry of Health. All animal experiments conformed to the guidelines of Directive 2010/63/EU of the European Parliament on the protection of animals used for scientific purposes.

### Cell culture
Human umbilical vein endothelial cells (HUVECs) were isolated from normal female human umbilical veins, which were collected from Qilu Hospital of Shandong University. HUVECs were cultured in Endothelial Cell Medium (ECM; ScienCell) supplemented with 5% heat-inactivated fetal bovine serum (ScienCell), 1% penicillin/streptomycin (ScienCell), and 1% endothelial cell growth supplement (ECGS; ScienCell). All experimental primary endothelial cells were used by passage 6. HEK293T cell line (CRL-11268), LLC cell line, B16 cell line, and MC38 cell line were purchased from ATCC. HEK293T, LLC, B16, MC38 cells were cultured in DMEM/high-glucose medium supplemented with 10% fetal bovine serum and antibiotics. All cells were cultured at 37 °C and 5% $CO_2$.

### Isolation of mouse retinal endothelial cells
The isolation of mouse retinal endothelial cells was performed as reported previously[69]. In brief, eyes were removed from euthanized mice, and retinas were collected in cold phosphate buffered saline (PBS). Retinas were then digested with collagenase type II (Sigma-Aldrich) and separated into single cells. Fluorescence-Activated Cell Sorting (FACS) was used to isolate CD31$^+$ cells and exclude CD45$^+$ cells for further study. MRECs were used by passage 3.

### Isolation and culture of mouse lung endothelial cells (MLECs)
Briefly, lungs were harvested, shredded, and digested with 0.1% collagenase in PBS for 45 min. The digest was homogenized by multiple passages through a 20-gauge needle and then filtered through a 70 μm tissue sieve. MLECs were isolated via immunoselection using CD31-conjugated (BD Pharmingen) magnetic beads (Invitrogen). When the plated cells reached confluency, a second immuno-isolation was performed using magnetic beads conjugated with intercellular adhesion molecule 2 (BD Pharmingen). Cells from passages 1 to 3 were used in this study.

### Hindlimb ischemia (HLI) model

Mice were anesthetized with 1.5% to 2% isoflurane vaporized in oxygen, administered with xylazine (6 mg/kg), placed on a 37 °C heating pad to maintain body temperature. A longitudinal 8 mm incision was made in the left inguinal crease along the femoral vessels visible through the skin. In brief, a portion of the femoral artery was exposed and ligated both proximally and distally using 6-0 silk sutures, and the vessels between the ligation sites were excised. We measured ischemic (left) and non-ischemic (right) limb blood flow ratios using a noninvasive laser Doppler imaging system (PeriCam PSI Z, Perimed) at baseline (day −1) and immediately after HLI (day 0), with subsequent imaging on days 7, 14, 21, and 28 post-HLI.

### Mouse retinal angiogenesis assay

Eyes from postnatal day 5 pups were fixed in 4% paraformaldehyde (PFA) for 2 h at 4 °C. Retinas were dissected and permeabilized overnight in PBS containing 1% Triton X-100 for 12 h at 4 °C. Then the permeabilized retinas were incubated in PBS containing 0.5% Triton X-100 and 5% BSA for 12 h at 4 °C. Retinas were stained with isolectin B4 - Alexa Fluor 594 (I21413, Invitrogen) for 12 h at 4 °C. Whole mounts were photographed using a fluorescence microscope.

### In vivo matrigel plug assay

A mixture containing 500 μL Matrigel (Corning), 50 ng/mL VEGF (Peprotech), and 30 U/mL heparin (Sigma-Aldrich) was subcutaneously injected into the dorsal surface of the mice, leading to the formation of a solid plug. Mice were sacrificed after 14 days and the Matrigel plugs were retrieved for analysis.

### Mouse skin wound healing model

Mice were anesthetized with 1.5% to 2% isoflurane in oxygen, and their back skin was wounded using a 5 mm biopsy punch without injuring the underlying muscle. Wounds were digitally photographed at days 0, 3, 5, and 7, and the wound diameters were measured using ImageJ to determine the wound closure rate.

### Mouse aortic sprouting assay

The thoracic aortas were isolated, trimmed of all extraneous tissues, and flushed with PBS via the lumen to remove all the blood. Aortic rings (1-mm long) were embedded in Matrigel (Corning) and cultured in Opti-MEM (Gibco) supplemented with 10% FBS and VEGF (20 ng/mL) in a humidified incubator at 37 °C, 5% $CO_2$ for approximately 6 days.

### Tumor transplantation model

Male CTR and dKO mice (8–10 weeks old) underwent allograft transplantations. Lewis lung carcinoma (LLC) cells ($4 \times 10^5$ cells), B16 tumor cells ($1 \times 10^6$ cells), and MC-38 cells ($8 \times 10^5$ cells) were suspended in 200 μL of Hank's balanced salt solution and injected subcutaneously into the murine flank. Tumor size was measured with digital calipers on days 8, 10, 12, 14, and 16 and calculated according to the following formula: $V = 0.52 \times L \times W^2$ ($V$, tumor volume; $L$, longest diameter of the tumor; $W$, perpendicular diameter of $L$)[70]. At the endpoint, the host mice were euthanized, and the tumors were dissected. After tumor weight was measured, they were fixed in 4% PFA and embedded in paraffin. Staining with the CD31 antibody was used for qualitative identification of ECs in the histological tissue sections.

### In vitro tube formation assay

Cells were serum-starved overnight in a medium containing 0.5% FBS. Matrigel (250 μL/well) was added to a 24-well plate, and $1 \times 10^5$ cells transfected with the indicated siRNAs or plasmids were plated on the Matrigel. After 4 h, tubes were visualized and images were captured using a phase-contrast inverted microscope. The length of the tubes and the number of branch points were quantified using ImageJ[71].

### Evans blue dye extravasation assay

Mice underwent Evans blue dye extravasation assay as described[72]. EB dye was obtained from Solarbio (Beijing, China). Male mice were injected intravenously with 20 mg/kg body weight of sterile EB dye. Mustard oil diluted to 5% in mineral oil was applied to the dorsal and ventral surfaces of the ear using a cotton swab, which was repeated after 15 min. Mice were anesthetized and photographs were taken 30 min after the injection of EB dye.

### Luciferase reporter assay

Luciferase reporter transfection and dual-luciferase assays were performed as follows. Briefly, HUVECs or HEK293T cells were seeded at $5 \times 10^4$ cells/well in 24-well plates and transfected using Lipofectamine 2000 (Invitrogen, USA) with 100 ng reporter vector carrying firefly luciferase (Promega, USA) and the indicated target sequences; an empty vector served as the control. The PRL-TK vector (carrying Renilla luciferase; Promega, USA) was co-transfected as an internal control. At 48 h after transfection, cells were lysed using passive lysis buffer (Promega, USA) and subjected to a luciferase assay according to the manufacturer's protocol.

### Chromatin immunoprecipitation (ChIP) assay

ChIP assays were performed using the SimpleChIP Enzymatic Chromatin IP kit (Magnetic Beads) (CST, #9005 S) according to the manufacturer's protocol. In brief, HUVECs with different treatments were incubated with 1% fresh paraformaldehyde at room temperature for 10 min to crosslink the histone/transcription factor complexes with DNA, followed by 0.1% glycine incubation at room temperature for 5 min. The nuclei pellets were digested with Micrococcal Nuclease at 37 °C for 20 min, followed by sonication. After centrifugation, the chromatin was immunoprecipitated with antibodies against Sp1 (1:50 dilution, Millipore, #07-645), Sp3 (1:50 dilution, Abcam, ab227856), and normal rabbit IgG (CST, #2729) overnight with gentle rotation. The protein/DNA complexes were immunoprecipitated by 30 μl ChIP grade protein G magnetic beads with rotation for 2 h at 4 °C, followed by three washes in low-salt buffer and one wash in high-salt buffer, and elution at 65 °C for 30 min. The eluted protein-DNA complexes were reversed with proteinase K at 65 °C for 2 h. The DNA was purified and then amplified by quantitative real-time PCR with the following primers targeted to the human NOTCH1 promoter (−219/+5), forward primer: 5′-GAAGTAGTCCCAGGCGCC-3′ and reverse primer: 5′-CTAGTGAGGCTCAGAGTCGA-3′. The ChIP-qPCR data were reported as a percentage of input, which can be calculated by the formula:

$$\% \text{ of input} = 2^{\wedge}((\text{Ct(input)} - \log2(\text{dilutionfactor})) - \text{Ct(ChIP)}) * 100\%$$

The input sample used was 10% of the DNA amount, and thus the dilution factor is 10.

### Co-immunoprecipitation

Endothelial cells or 293 T cells were washed twice with cold PBS after various treatments and harvested in IP lysis buffer (Invitrogen). Cell lysates were then centrifuged at 12000 $g$ for 20 min at 4 °C. Supernatants were immediately prepared for immunoprecipitation by incubating protein A/G magnetic beads (Bimake) for 2 h at 4 °C to preclear before incubating with the first protein-specific antiserum. Then, 20 μL of protein A/G magnetic beads was added to the antibody-lysate mix and incubated overnight at 4 °C. Immune complexes were collected and washed 5 times with lysis buffer. After a final wash, the supernatant was aspirated and discarded, then the precipitated proteins were eluted from the beads by resuspending in 2 × SDS-PAGE loading buffer and boiling for 5 min. The resultant materials from immunoprecipitation or cell lysates underwent western blot analysis.

## Extraction of cytoplasmic and nuclear proteins and western blot analysis

Whole cell lysates were extracted with RIPA buffer containing Complete Protease Inhibitor Cocktail Tablets and phosphorylase inhibitor (CWBIO). Nuclear and cytoplasmic extracts were prepared from HUVECs by using NE-PER Nuclear and Cytoplasmic Extraction Reagents (ThermoFisher), supplemented with Complete Protease Inhibitor Cocktail Tablets and phosphorylase inhibitor. Protein samples were separated by 10% SDS-PAGE and transferred onto PVDF membrane (Millipore). After incubation with horseradish peroxidase–conjugated secondary antibodies, proteins were visualized with Amersham ImageQuant 680 (Cytiva). Band intensity was quantified by scanning densitometry of the autoradiogram with NIH ImageJ.

## Immunofluorescence staining

Human muscle samples and other mouse samples were fixed in 4% PFA at 4 °C overnight, then embedded in optimal cutting temperature (OCT) compound (Tissue-Tek), and cut into 5 μm sections for staining. Sections were blocked in 10% goat serum for 1 h at 26 °C, then incubated with primary antibody at 4 °C overnight. After washing with PBS, the sections were incubated with Alexa Fluor-labeled secondary antibodies at room temperature for 1 h. Finally, the nuclei were stained with 4,6′-diamidino-2-phenylindole (DAPI) for 5 min at room temperature, and immunofluorescence was analyzed using a fluorescent microscope.

Cells were cultured in 24-well chamber slides and transfected with the indicated siRNAs or treated with other reagents. Cells were washed with PBS, fixed using 4% PFA, permeabilized with 0.1% Triton X-100 in PBS, blocked with 10% goat serum, and incubated with the corresponding antibodies overnight, at 4 °C. After PBS wash, cells were incubated with Alexa fluor-conjugated secondary antibody. An aqueous Fluoroshield mounting medium with DAPI (Abcam) was used to cover the cells.

## Human samples

Human tissue biopsies from gastrocnemius were obtained from 10 patients (7 male and 3 female) in Qilu Hospital of Shandong University. Five patients with CLI with an average of 56 were included in this study. 5 patients without CLI with an average of 59 were included in this study. All patients gave written informed consent to participate in the study. All procedures involving human samples were approved by the Ethical Committee of Qilu Hospital of Shandong University. All relevant ethical regulations were followed in this study.

## Reporting summary

Further information on research design is available in the Nature Portfolio Reporting Summary linked to this article.

## Data availability

Data supporting the findings of this study are available within the article and its Supplementary Information files. Source data are provided with this paper.

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

## Acknowledgements

The authors thank Prof. Yulong He from Soochow University for providing the VE-CAD-CreER$^{T2}$ mice. This work was supported by the Natural Science Foundation for Distinguished Young Scholars of Shandong Province (ZR2020JQ30 to W.Z.), the National Natural Science Foundation of China (81970198 to W.Z., 81970366 to J.Y. and 82030051 to Y.Z.), and National Key Research and Development Project of China (2021YFA1301102 to Q.Z.).

## Author contributions

H.L., P.Y., X.M., X.J., S.L., C.M., S.P., Q.Z., J.Y., F.X., C.Z., Y.Z., and W.Z. conceived and designed the research; H.L., C.M., P.Y., and X.M. performed experiments; H.L., X.M., and P.Y. analyzed data; H.L., P.Y., X.M., X.J., S.L., C.M., S.P., Q.Z., J.Y., F.X., C.Z., Y.Z., and W.Z. interpreted results of experiments; H.L., P.Y., and W.Z. drafted the manuscript; H.L., P.Y., X.M., and W.Z. edited and revised manuscript. H.L., P.Y., X.M., X.J., S.L., C.M., S.P., Q.Z., J.Y., F.X., C.Z., Y.Z., and W.Z. approved the final version of the manuscript.

## Competing interests

The authors declare no competing interests.
