## [Peer Review File · Nature Communications]

Angiotensin-Converting Enzyme Inhibitor Promotes Angiogenesis through Sp1/Sp3-Mediated Inhibition of Notch Signaling in male miceREVIEWER COMMENTS

Reviewer #1 (Remarks to the Author):

The authors reveal the role of endothelial Sp1/Sp3 in tissue angiogenesis. They demonstrate that genetic deletion of Sp1/Sp3 significantly impairs angiogenesis during postnatal development by activating Notch-1 signaling and in turn inhibiting VEGFR2 signaling. Moreover, this study elucidates the mechanism of proangiogenic effect of ACE-I by focusing ubiquitination and acetylation of Sp1/Sp3. Their approach is interesting, however, some critical issues are still unclear.

1. This study suggests that ACE-I has the proangiogenic effect by activating the deubiquitinating enzyme, USP7 as well as decreasing acetylation of Sp1/Sp3. However, it remains unclear how ACE-I exerts these effects in endothelial cells. By blocking ACE? or by blocking angiotensin-II/angiotensin-II receptor axis? The authors should elucidate this pharmacological mechanism.
2. This study demonstrates that Sp1/Sp3-dKO mice show the reduced tumor growth, tumor mass in several xenograft models. The effects of ACE-I on these xenografted tumors should be clarified.
3. According to the underlying mechanism of ACE-I on USP7-SP1/Sp3-Notch1 signaling, ACE-I is supposed to exacerbate tumor angiogenesis. I concern that this may contradict with large number of reports regarding the inhibitory effect of ACE-I against tumor angiogenesis.

Reviewer #2 (Remarks to the Author):

This manuscript describes a variety of genetic and pharmacological factors (ACE inhibitors which are frequently prescribed to lower blood pressure) which all are interconnected to regulate angiogenic sprouting in the neonatal mouse retina models, hindlimb ischemia, wound healing, several mouse tumor models, Matrigel plug, and cell culture models (tube formation, spheroid, aortic ring). Moreover, for some of these genes relative expression strength association studies were performed using patient material (limb ischemia).

More specifically, the data indicate that ACE inhibition may lead to increased angiogenesis under certain conditions and that this requires the presence of sp1/sp3 transcription factors. These bind to the Notch1 promoter and activate its transcription. The authors propose the ACE inhibitors activate USP7 which de-ubiquitylates HDAC1. This leads to Notch1 transcription which subsequent repression of VEGFR2 expression.

In summary, the data are obviously very interesting and have some translational impact. There are novel findings about signaling in endothelial cells for the regulation of angiogenesis.

Major points:

1) this manuscript is difficult to digest. It addresses a large variety of genetic factors in various models and the story is hard to follow. It might be easier to tell it like this: ACE inhibition and the angiogenesis phenotype. What happens downstream? USP7, HDAC1 and sp1/sp3. What finally regulates sprouting: Notch and VEGFR2 (which is very well known).

As it is, the manuscript does not read like a typical Nature paper.

2) Little is known about the pro-angiogenic roles of ACE inhibitors. The drugs have been described to millions of people and vessel destabilizing effects are not a problem. Indeed, ACE inhibitors are somehow vessel protective. The three cited papers show better blood perfusion in hindlimb ischemia models upon chronic ACE inhibitor treatment. However, this effect is usually due to arteriogenesis and not sprouting angiogenesis. Although somehow similar, both processes need to be separated from each other.

In addition, the authors need to justify the dose of ACE inhibitor injected into the eyes of newborn mice. Is this dose comparable with tissue or plasma concentration of these drugs when giving at a normal dose p.o.

3) To justify whether ACE inhibition requires sp1/sp3 to induce retinal angiogenesis the authors use the EC-specific KO mice. This gives an impressive result. However, it could well be that sp1/sp3 deletion would simply block all pro-angiogenic signaling events. Therefore, it would be informative to whether VEGF, angiopoietin or similar injections would rescue the sp1/sp3 phenotype.

4) The authors generated EC-specific sp1/sp3 knockout mice which have an impressive vascular phenotype. However, any other information about these mice is missing. What is the recombination rate in ECs or how strong is the gene depletion? Do the mice show any abnormalities. What about survival rate, body weight? Is there already a reduced vessel number in limb muscle before ligating the artery? The same for skin and the wound healing model. Are there additional vascular defects (permeability, inflammation)?

5) Does EC-specific sp1/sp3 deletion lead to increased hypoxia in the tumor models?

6) The link between ACE inhibitor and USP7 come somehow out of the blue. What is the rationale for investigating this class of proteins? How could this drug affect the activity of this de-ubiquitinating enzyme?

7) Both USP7 and HDAC1 should influence the activity or abundance of numerous genes and proteins. Therefore, one is wondering why the effect on Notch1 is so pronounced. The authors need to address whether there is a vast change in global gene transcription or whether only very few specific genes are altered.

8) The Discussion is too long. Please focus.

Reviewer #3 (Remarks to the Author):

This interesting manuscript uncovers critical downstream signalings of ACEI in angiogenesis. The authors show that ACEI regulates the activity of endothelial cells via the SP1/SP3-Notch-VEGFR2 axis upon the regulation of USP7-mediated ubiquitination and HDAC1-mediated acetylation. Overall, this paper is a great job and the findings potentially add significant knowledge to the pharmacological effect of ACEI in angiogenesis and correlated diseases. However, I have following concerns.

1. The 'Introduction' part needs to be modified and reorganized, especially the summary about the reported effect of ACEI on Notch and VEGFR signaling. Also, why the authors hypothesize SP1/SP3 could act as key transcription factors at downstream of ACEI. Are there any clues?

2. In figure 1, the authors showed the change of expression pattern of SP1/SP3 in tumor angiogenesis specimens. Can the similar results also be observed in samples of physiological angiogenesis, either in clinical or animal samples? How about the Notch and VEGFR signaling in these pathological/physiological angiogenesis samples? Moreover, quantification of the staining signals is required.

3. In figure 2B, dual luciferase assay showing HEY1/HES1-mediated repression of the VEGFR2 promoter is required, which is for a better link between NOTCH and VEGFR2.

4. The authors showed SP1 and SP3 work in a same transcription complex interactively, but they also function independently. Further explanation, at least deeper discussion, is required to illustrate their working machinery.

5. A MG132 rescue assay is required to prove the USP7-mediated ubiquitination on SP1 and SP3's stability.

6. It is important to verify whether ACEI affects the classic angiogenic factors such as VEGFA and other angiogenic growth factors, and the induction of HIF family. Also, it is necessary to identify whether ACEI directly affects the activation of VEGFR2 independent of NOTCH.

Reviewer #4 (Remarks to the Author):

This research has certain novelty. The work supports the conclusions and claims. The methodology is sound. The structure of the paper is reasonable.

But the part of "introduction" should be improved. The relevance between Sp1/Sp3 and Notch needs to be explained. The relevance between ACEI and Notch needs to be explained.

We thank the reviewers for their careful readings. We are delighted to know that **“Their approach is interesting.”** (reviewer #1), **“In summary, the data are obviously very interesting and have some translational impact.”** (reviewer #2), **“Overall, this paper is a great job and the findings potentially add significant knowledge to the pharmacological effect of ACEI in angiogenesis and correlated diseases.”** (reviewer #3), and **“This research has certain novelty. The work supports the conclusions and claims. The methodology is sound. The structure of the paper is reasonable.”** (reviewer #4). Thanks to these positive comments, we are extremely encouraged to revise the manuscript by performing all additional experiments. We found the reviewers’ comments very helpful, which guided our revisions resulting in this much improved manuscript. Because there are limited changes to this manuscript based on the review, changes in text are noted by the **red color texts**.

Point-by-point response to the reviewers’ comments

Reviewer #1 (Remarks to the Author):

R1.1. This study suggests that ACE-I has the proangiogenic effect by activating the deubiquitinating enzyme, USP7 as well as decreasing acetylation of Sp1/Sp3. However, it remains unclear how ACE-I exerts these effects in endothelial cells. By blocking ACE? or by blocking angiotensin-II/angiotensin-II receptor axis? The authors should elucidate this pharmacological mechanism.

Response: We thank the reviewer for this comment. In this study and our another study, we found that ACEI could directly act on endothelial cells independent of angiotensin-II and bradykinin. To answer the reviewer’s question more comprehensively, we respectfully submit that our findings herein suggest ACEI phosphorylating USP7 via protein kinase CK2. Phosphorylation of USP7 at serine 18 by CK2 is required for USP7 stability.^{1,2} Additionally, CK2 inhibition leads to decreased angiogenesis.³⁻⁵ Thus, we concluded that ACEI regulates USP7 by directly acting on endothelial cells via CK2-mediated phosphorylation.

As shown in **Fig. S9A**, ACEI increased the phosphorylation level of USP7 in HUVECs, which was inhibited by CX-4945 (inhibitor against catalytic CK2 α and CK2 α' subunits). Further, CX-4945 attenuated the increased protein level of Sp1/Sp3 induced by ACEI (**Fig. S9A**) and blocked USP7 translocating into the nucleus (**Fig. S9A and B**).

R1.2. This study demonstrates that Sp1/Sp3-dKO mice show the reduced tumor growth, tumor mass in several xenograft models. The effects of ACE-I on these xenografted tumors should be clarified.

Response: We thank the reviewer for this comment. As suggested, we have added a new supplementary **Fig. S4** that presents our data demonstrating the effect of ACEI on the xenografted tumors. LLC, B16 and MC38 cells were injected subcutaneously into the flanks of either vehicle or ACEI-treated C57BL/6J mice. After 16 days, ACEI-treated mice exhibited the similar tumor growth, tumor weight and blood vessels density compared with control group.

R1.3. According to the underlying mechanism of ACE-I on USP7-SP1/Sp3-Notch1 signaling, ACE-I is supposed to exacerbate tumor angiogenesis. I concern that this may contradict with large number of reports regarding the inhibitory effect of ACE-I against tumor angiogenesis.

Response: We thank the reviewer for this comment, which is similar to R1.2. According to our new data in **Fig. S4**, we found that ACEI had no effect on xenografted tumors. The effects of ACEI on tumor growth were controversial in previous reports especially in clinical study.⁶⁻¹¹ The factors influence tumor angiogenesis are various: tumor cells, immune cells, vascular cells and the tumor microenvironment. And there are multiple angiogenesis-associated molecules such as vascular endothelial growth factor (VEGF), epidermal growth factor (EGF), hepatocyte (HGF), fibroblast growth factor (FGF) and angiopoietin (Ang). Thus, our findings on ACEI-Sp1/Sp3-Notch1-VEGFR2 only focused on part of the angiogenic mechanism in endothelial cells, which could not fully account for the complicated pharmacological effect of ACEI on tumor

angiogenesis *in vivo*.

Reviewer #2 (Remarks to the Author):

R2.1. this manuscript is difficult to digest. It addresses a large variety of genetic factors in various models and the story is hard to follow. It might be easier to tell it like this: ACE inhibition and the angiogenesis phenotype. What happens downstream? USP7, HDAC1 and sp1/sp3. What finally regulates sprouting: Notch and VEGFR2 (which is very well known). As it is, the manuscript does not read like a typical Nature paper.

Response: We thank the reviewer for this comment. As suggested, we have revised the manuscript. Firstly, we introduced the angiogenesis phenotype in endothelial Sp1/Sp3-deletion mice and ACEI-treated mice. Then, we explained the molecular mechanisms between ACEI and Sp1/Sp3. Finally, we explored how endothelial Sp1/Sp3 regulate angiogenesis: Notch1-VEGFR2 signaling.

R2.2. Little is known about the pro-angiogenic roles of ACE inhibitors. The drugs have been described to millions of people and vessel destabilizing effects are not a problem. Indeed, ACE inhibitors are somehow vessel protective. The three cited papers show better blood perfusion in hindlimb ischemia models upon chronic ACE inhibitor treatment. However, this effect is usually due to arteriogenesis and not sprouting angiogenesis. Although somehow similar, both processes need to be separated from each other.

In addition, the authors need to justify the dose of ACE inhibitor injected into the eyes of newborn mice. Is this dose comparable with tissue or plasma concentration of these drugs when giving at a normal dose p.o.

Response: We thank the reviewer for these comments.

With regard to the first comment, thank you for the opportunity to clarify the arteriogenesis and angiogenesis in hindlimb ischemia model. Surgical ligation of the

femoral artery at a specific site leads to arteriogenesis in femoral collaterals and angiogenesis in distal ischemic muscles.¹²⁻¹⁴ Blood flow recovery is thought to be dependent on arteriogenesis, with most new arteries forming above the site of ligation. However, some flow recovery is undoubtedly attributable to angiogenesis in the ischemic part of the limb.¹⁴ Ischemic angiogenesis is induced in the gastrocnemius and can be quantified on gastrocnemius sections. The amount of ischemia-induced angiogenesis can be evaluated by IHC staining with anti-CD31 antibody compared with the non-ligated side of mice. Quantification of arteriogenesis is carried out on femoral collaterals in the center of the semitendinosus.¹² For these reasons, we chose hindlimb ischemia model and IHC staining on gastrocnemius sections for evaluating pathological angiogenesis.

With regard to the second comment, we apologized for the error in the discussion. We have fully described the methods in the supplementary materials. We tried to administrate the pups with ACEI by intraocular injection but failed to control the injection volume. Therefore, ACEI was administered by intraperitoneal injection to each pup.

“For ACEI treatment, 8-week-old mice were injected intraperitoneally with an ACEI (perindopril, 3 mg/kg/d) for 28 days in hindlimb ischemia model or for 14 days in a Matrigel plug assay. In the postnatal retinal angiogenesis model, ACEI (perindopril, 1 mg/mL, 10 μ L) was administered by intraperitoneal injection to each pup from P2 to P4.”

R2.3. To justify whether ACE inhibition requires sp1/sp3 to induce retinal angiogenesis the authors use the EC-specific KO mice. This gives an impressive result. However, it could well be that sp1/sp3 deletion would simply block all pro-angiogenic signaling events. Therefore, it would be informative to whether VEGF, angiopoietin or similar injections would rescue the sp1/sp3 phenotype.

Response: We thank the reviewer for this comment. To answer the reviewer's question, we have added a new supplementary Fig. 11 that presents our data demonstrating that deletion of Sp1/Sp3 cannot block all proangiogenic signaling. VEGF and angiopoietin-1 (Ang-1) are essential factors to promote angiogenesis through regulation of various signaling events in endothelial cells.¹⁵⁻¹⁷ As shown in **Fig. S12A**, we isolated mouse lung endothelial cells (MLECs) from CTR and dKO mice. *In vitro* capillary tube formation assay showed that the combined action of VEGF and Ang1 promoted angiogenesis in dKO MLECs (**Fig. S12A**). Corresponding to the tube formation assay, the aortic rings from dKO mice also responded to VEGF and Ang1 *ex vivo* (**Fig. S12B**).

R2.4. The authors generated EC-specific sp1/sp3 knockout mice which have an impressive vascular phenotype. However, any other information about these mice is missing. What is the recombination rate in ECs or how strong is the gene depletion? Do the mice show any abnormalities. What about survival rate, body weight? Is there already a reduced vessel number in limb muscle before ligating the artery? The same for skin and the wound healing model. Are there additional vascular defects (permeability, inflammation)?

Response: We thank the reviewer for these comments.

With regard to the first comment, we verified the Sp1/Sp3 deletion in retinal endothelial cells (Figure 7A).

With regard to the second comment, we exhibited the survival rate, body weight and blood pressures in another manuscript under review. EC-specific Sp1/Sp3 knockout mice resulted in embryonic lethality, thus we generated tamoxifen-induced Sp1/Sp3^{ECKO} mice. dKO mice showed decreased survival rate and body weight and increased blood pressures compared with CTR mice.

With regard to the third comment, we have measured the vessel density in gastrocnemius before ligating the femoral artery. As shown below, there was no difference between CTR and dKO mice.

With regard to the fourth comment, we have detected the vascular permeability by injecting the CTR and dKO mice with Evans blue (EB) dye. EB binds tightly to plasma proteins and is normally retained in the vascular space, its extravasation indicating protein leakage into the interstitial space. As shown below, dKO mice exhibited almost the same EB-albumin extravasation in the paw, subcutaneous gelatin, and perivascular adipose tissue.

With regard to the fifth comment, we have detected the serum IL-1 β and TNF- α . As shown below, there was no difference between CTR and dKO mice.

R2.5. Does EC-specific sp1/sp3 deletion lead to increased hypoxia in the tumor models?

Response: We thank the reviewer for this comment. To answer the reviewer's question, we have added new data in supplementary **Fig. S1E**, **Fig. S2E** and **Fig. S3E**. In LLC, B16 and MC38 xenograft models, reduced vessel density following endothelial Sp1/Sp3 deletion resulted in larger areas of hypoxia labeled by HIF1- α .

R2.6. The link between ACE inhibitor and USP7 come somehow out of the blue. What is the rationale for investigating this class of proteins? How could this drug affect the activity of this de-ubiquitinating enzyme?

Response: We thank the reviewer for this comment, which is similar to R1.1. In this study and our another study, we found that ACEI could directly act on endothelial cells independent of angiotensin-II and bradykinin and decrease the ubiquitination level of Sp1/Sp3. To answer the reviewer's question more comprehensively, we respectfully submit that our findings herein suggest captopril phosphorylating USP7 via protein kinase CK2. Phosphorylation of USP7 at serine 18 by CK2 is required for USP7 stability.^{1,2} Additionally, CK2 inhibition leads to decreased angiogenesis.³⁻⁵ Thus, we concluded that ACEI regulates USP7 by directly acting on endothelial cells via CK2-mediated phosphorylation. As shown in **Fig. S9A**, ACEI increased the phosphorylation level of USP7 in HUVECs, which was inhibited by CX-4945 (inhibitor against catalytic CK2 α and CK2 α' subunits). Further, CX-4945 attenuated the increased protein level of

Sp1/Sp3 induced by ACEI (**Fig. S9A**) and blocked USP7 translocating into the nucleus (**Fig. S9A and B**).

R2.7. Both USP7 and HDAC1 should influence the activity or abundance of numerous genes and proteins. Therefore, one is wondering why the effect on Notch1 is so pronounced. The authors need to address whether there is a vast change in global gene transcription or whether only very few specific genes are altered.

Response: We thank the reviewer for this comment. In this study, we have used transcriptomic analysis of mouse lung endothelial cells (MLECs) from CTR and dKO mice and identified 2683 genes with altered expression upon Sp1/Sp3 deletion (1591 upregulated and 1092 downregulated, $p < 0.05$ and $|\text{Log2FoldChange}| > 1.5$). According to the differentially expressed genes, we decided to explore the Notch1-VEGFR2 signaling. The RNA-seq data generated in this study have been deposited in the NCBI Gene Expression Omnibus (GEO) database under accession code GSE GSE206586.

R2.8. The Discussion is too long. Please focus.

Response: We thank the reviewer for this comment. We have rewritten the discussion to focus more on the main message and removed the unnecessary text in the revised manuscript.

Reviewer #3 (Remarks to the Author):

R3.1. The 'Introduction' part needs to be modified and reorganized, especially the summary about the reported effect of ACEI on Notch and VEGFR signaling. Also, why the authors hypothesize SP1/SP3 could act as key transcription factors at downstream of ACEI. Are there any clues?

Response: We thank the reviewer for these comments.

With regard to the first comment, we have modified and reorganized the introduction, especially the summary about the effect of ACEI on Notch and VEGFR2 signaling in the revised manuscript (Page 4, Lines 60 - 68).

With regard to the second comment, thank you for the opportunity to clarify the reason for the hypothesis. In another study under submission, we found that Sp1 and Sp3 deletion abolished the captopril (a type of ACEI)-induced alleviation of hypertension, endothelial dysfunction and the anti-apoptotic effects; Sp1 and Sp3 have an indispensable role in the success of captopril treatment in the endothelium. Additionally, Sp1/Sp3 have been demonstrated as crucial transcription factors regulating angiogenesis.¹⁸ Thus, we hypothesized that Sp1/Sp3 could act as key transcription factors at downstream of ACEI.

R3.2. In figure 1, the authors showed the change of expression pattern of SP1/SP3 in tumor angiogenesis specimens. Can the similar results also be observed in samples of physiological angiogenesis, either in clinical or animal samples? How about the Notch and VEGFR signaling in these pathological/physiological angiogenesis samples? Moreover, quantification of the staining signals is required.

Response: We thank the reviewer for these comments.

With regard to the first comment, we have detected the Sp1/Sp3 expressions in the new vessels after ligation of femoral artery. As shown below, we found increased Sp1/Sp3 expressions in CD31-labeled vessels of gastrocnemius compared with sham hindlimb.

With regard to the second comment, we have added a new supplementary **Fig. S10** that presents our data demonstrating the expressions of VEGFR2 and NOTCH1 in the CD31-labeled vessels of the LLC xenografted tumors. In the LLC xenografted tumors of dKO mice, the expression of VEGFR2 was downregulated and NOTCH1 was upregulated compared with CTR mice.

With regard to the third comment, we have carried out densitometric quantification using the ImageJ software for all the immunofluorescence staining data in the revised manuscript.

R3.3. In figure 2B, dual luciferase assay showing HEY1/HES1-mediated repression of the VEGFR2 promoter is required, which is for a better link between NOTCH and VEGFR2.

Response: We thank the reviewer for this comment. The effects of Notch1 on VEGFR2 have been fully demonstrated in previous studies.¹⁹⁻²¹ To answer the reviewer's question, we conducted dual luciferase assay in HUVECs. It was demonstrated that knockdown of HEY1/HES1 by siRNA attenuated Ad-NICD-induced repression of the VEGFR2 promoter (**Figure. 8F**).

R3.4. The authors showed SP1 and SP3 work in a same transcription complex interactively, but they also function independently. Further explanation, at least

deeper discussion, is required to illustrate their working machinery.

Response: We thank the reviewer for this comment. We have demonstrated the separate function of Sp1 and Sp3 in the last part of the results. To explore the distinct roles of Sp1 and Sp3 in Notch1 signaling *in vitro*, HUVECs were treated with Sp1 or Sp3 siRNA. The knockdown resulted in the upregulation of protein levels of Notch1, NICD, and DLL4 and the mRNA levels of NOTCH1, HES1, HEY1, and DLL4 (**Figure 8A and 8B**). We then performed siRNA-mediated Sp1 knockdown in Sp3 overexpression (OE) ECs or Sp3 knockdown in Sp1 OE ECs. Sp1 knockdown in Sp3 OE ECs, led to a strong activation of Notch1 signaling (**Fig. S15A and B**) and increased mRNA levels (**Fig. S15C**). However, Sp3 siRNA-treated Sp1 OE ECs did not activate Notch1 signaling. These findings suggest that Sp3-mediated inactivation of Notch1 signaling depends on the presence of Sp1.

A simultaneous knockdown of expression using Sp1 and Sp3 siRNA induced further upregulation of Notch1 signaling, indicating a cooperative relationship between Sp1 and Sp3 (**Figure 8A and 8B**). Sp1/Sp3 modulate transcription by binding to GC-box sequences in the regulatory regions of its target genes. Several GC-box motifs are proximal to the NOTCH1 transcription start site (TSS). Chromatin immunoprecipitation (ChIP) was used to verify whether Sp1 and Sp3 interacted with the NOTCH1 promoter. Nuclear extracts from HUVECs were incubated with anti-Sp1 and anti-Sp3 antibodies or IgG (negative control). The ChIP primer for NOTCH1 was designed to amplify promoter regions containing putative binding sites for Sp1 and Sp3. We observed that both Sp1 and Sp3 could bind to the NOTCH1 promoter (**Figure 8C**).

To determine the independent roles of endothelial Sp1 and Sp3 in promoting angiogenesis, we analyzed retinal vessel development at P5 in *VE-CAD-CreER^{T2+}/Sp1^{fl/fl}* (Sp1^{ECKO}) and *VE-CAD-CreER^{T2+}/Sp3^{fl/fl}* (Sp3^{ECKO}) mice. In accordance with the *in vitro* data, retinas from Sp1^{ECKO} mice and Sp3^{ECKO} mice showed a mild delay in vasculature development phenotype compared with that of dKO mice. Furthermore, retinas from Sp1^{ECKO} mice exhibited more severely impaired angiogenesis than Sp3^{ECKO}, indicating the prior role of Sp1 in angiogenesis to Sp3 in ECs (**Figure 8G**).

R3.5. A MG132 rescue assay is required to prove the USP7-mediated ubiquitination on SP1 and SP3's stability.

Response: We thank the reviewer for this comment. We have added a new Supplementary **Fig. S6** that presents an MG132 rescue assay to prove the USP7-mediated ubiquitination on Sp1 and Sp3's stability.

R3.6. It is important to verify whether ACEI affects the classic angiogenic factors such as VEGFA and other angiogenic growth factors, and the induction of HIF family. Also, it is necessary to identify whether ACEI directly affects the activation of VEGFR2 independent of NOTCH.

Response: We thank the reviewer for these comments.

With regard to the first comment, we have added a new Supplementary **Fig. S14** that presents our data verifying that ACEI did not affect the expressions of VEGFA and HIF-1 α .

With regard to the second comment, we have added a new Supplementary **Fig. S13A**. In HUVECs pretreated with ACEI, overexpression of NICD by Ad-NICD

inhibited the VEGF-induced phosphorylation of VEGFR2, PLC γ , and ERK1/2 considerably. Thus, ACEI could not directly affect the activation of VEGFR2.

Reviewer #4 (Remarks to the Author):

This research has certain novelty. The work supports the conclusions and claims.

The methodology is sound. The structure of the paper is reasonable.

But the part of “introduction” should be improved. The relevance between Sp1/Sp3 and Notch needs to be explained. The relevance between ACEI and Notch needs to be explained.

Response: We thank the review for these positive comments. As suggested, we have explained the relevance between Sp1/Sp3 and Notch (Page 3, Lines 50 - 52), and the relevance between ACEI and Notch in the revised manuscript (Page 4, Lines 60 - 68) as follows:

References

- 1 Khoronenkova, S. V. *et al.* ATM-dependent downregulation of USP7/HAUSP by PPM1G activates p53 response to DNA damage. *Molecular cell* **45**, 801-813, doi:10.1016/j.molcel.2012.01.021 (2012).
- 2 Fernández-Montalván, A. *et al.* Biochemical characterization of USP7 reveals post-translational modification sites and structural requirements for substrate processing and subcellular localization. *The FEBS journal* **274**, 4256-4270, doi:10.1111/j.1742-4658.2007.05952.x (2007).
- 3 Feng, D. *et al.* Protein kinase CK2 is a regulator of angiogenesis in endometriotic lesions. *Angiogenesis* **15**, 243-252, doi:10.1007/s10456-012-9256-2 (2012).
- 4 Schmitt, B. M. *et al.* Protein Kinase CK2 Regulates Nerve/Glial Antigen (NG)2-Mediated Angiogenic Activity of Human Pericytes. *Cells* **9**, doi:10.3390/cells9061546 (2020).
- 5 Benavent Acero, F. *et al.* CIGB-300, an anti-CK2 peptide, inhibits angiogenesis, tumor cell invasion and metastasis in lung cancer models. *Lung cancer (Amsterdam, Netherlands)* **107**, 14-21, doi:10.1016/j.lungcan.2016.05.026 (2017).
- 6 Zhang, S. *et al.* Angiotensin-converting enzyme inhibitors have adverse effects in anti-angiogenesis therapy for hepatocellular carcinoma. *Cancer letters* **501**, 147-161, doi:10.1016/j.canlet.2020.12.031 (2021).
- 7 Khanna, P. *et al.* ACE2 abrogates tumor resistance to VEGFR inhibitors suggesting angiotensin-(1-7) as a therapy for clear cell renal cell carcinoma. *Science translational medicine* **13**, doi:10.1126/scitranslmed.abc0170 (2021).

- 8 Yoon, C., Yang, H. S., Jeon, I., Chang, Y. & Park, S. M. Use of angiotensin-converting-enzyme inhibitors or angiotensin-receptor blockers and cancer risk: a meta-analysis of observational studies. *CMAJ : Canadian Medical Association journal = journal de l'Association medicale canadienne* **183**, E1073-1084, doi:10.1503/cmaj.101497 (2011).
- 9 Okwan-Duodu, D., Landry, J., Shen, X. Z. & Diaz, R. Angiotensin-converting enzyme and the tumor microenvironment: mechanisms beyond angiogenesis. *American journal of physiology. Regulatory, integrative and comparative physiology* **305**, R205-215, doi:10.1152/ajpregu.00544.2012 (2013).
- 10 Volpert, O. V. *et al.* Captopril inhibits angiogenesis and slows the growth of experimental tumors in rats. *The Journal of clinical investigation* **98**, 671-679, doi:10.1172/jci118838 (1996).
- 11 Copland, E. *et al.* Antihypertensive treatment and risk of cancer: an individual participant data meta-analysis. *The Lancet. Oncology* **22**, 558-570, doi:10.1016/s1470-2045(21)00033-4 (2021).
- 12 Limbourg, A. *et al.* Evaluation of postnatal arteriogenesis and angiogenesis in a mouse model of hind-limb ischemia. *Nature protocols* **4**, 1737-1746, doi:10.1038/nprot.2009.185 (2009).
- 13 Couffignal, T. *et al.* Mouse model of angiogenesis. *The American journal of pathology* **152**, 1667-1679 (1998).
- 14 Simons, M. *et al.* State-of-the-Art Methods for Evaluation of Angiogenesis and Tissue Vascularization: A Scientific Statement From the American Heart Association. *Circulation research* **116**, e99-132, doi:10.1161/res.0000000000000054 (2015).
- 15 Koblizek, T. I., Weiss, C., Yancopoulos, G. D., Deutsch, U. & Risau, W. Angiopoietin-1 induces sprouting angiogenesis in vitro. *Current biology : CB* **8**, 529-532, doi:10.1016/s0960-9822(98)70205-2 (1998).
- 16 Tao, Z. *et al.* Coexpression of VEGF and angiopoietin-1 promotes angiogenesis and cardiomyocyte proliferation reduces apoptosis in porcine myocardial infarction (MI) heart. *Proceedings of the National Academy of Sciences of the United States of America* **108**, 2064-2069, doi:10.1073/pnas.1018925108 (2011).
- 17 Chidiac, R. *et al.* Comparative Phosphoproteomics Analysis of VEGF and Angiopoietin-1 Signaling Reveals ZO-1 as a Critical Regulator of Endothelial Cell Proliferation. *Molecular & cellular proteomics : MCP* **15**, 1511-1525, doi:10.1074/mcp.M115.053298 (2016).
- 18 Pagès, G. & Pouyssegur, J. Transcriptional regulation of the Vascular Endothelial Growth Factor gene--a concert of activating factors. *Cardiovascular research* **65**, 564-573, doi:10.1016/j.cardiores.2004.09.032 (2005).
- 19 Taylor, K. L., Henderson, A. M. & Hughes, C. C. Notch activation during endothelial cell network formation in vitro targets the basic HLH transcription factor HESR-1 and downregulates VEGFR-2/KDR expression. *Microvascular research* **64**, 372-383, doi:10.1006/mvre.2002.2443 (2002).
- 20 Holderfield, M. T. *et al.* HESR1/CHF2 suppresses VEGFR2 transcription independent of binding to E-boxes. *Biochemical and biophysical research communications* **346**, 637-648, doi:10.1016/j.bbrc.2006.05.177 (2006).
- 21 Hultgren, N. W. *et al.* Slug regulates the Dll4-Notch-VEGFR2 axis to control endothelial cell activation and angiogenesis. *Nature communications* **11**, 5400, doi:10.1038/s41467-

020-18633-z (2020).

REVIEWER COMMENTS

Reviewer #1 (Remarks to the Author):

I appreciate the authors to correctly answer all my concerns.

Reviewer #2 (Remarks to the Author):

The authors have substantially revised and thereby improved their manuscript. However, some points were not addressed properly so I would like to spell them out again.

1) While the dose and the route of application of ACE inhibitors have been clarified, it still remains open why this particular dose was chosen. Is this comparable with plasma or tissue concentrations detected in humans treated with ACEi?

2) The authors now provide some more information about the sp1/sp3 EC-KO mice (vessel density in muscle, Evans blue permeability, IL1b, TNFa). This information should be given to all readers and not only the reviewer. Please add it to the suppl. figures.

3) The whole manuscript is poorly written and not sufficient for publication. The introduction is a simple description of all of the involved molecules. No connections, no hypotheses, no rationale.. The different parts of the Results chapter are poorly linked with each other. The Discussion is a chaotic mixture of experimental details, superficial associations with some clinical data, and many more.

Reviewer #3 (Remarks to the Author):

The authors have addressed most of the questions properly. However, there are a few concerns for minor revision.

1. The fluorescent signal in Figure S10 is too weak. Improved images are required.
2. a summary of the points in R 3.4 could be added in the discussion part of the manuscript.

We thank the reviewers for their thorough evaluation of this manuscript and the opportunity to strengthen our findings and conclusions. Thanks to these comments, we are extremely encouraged to revise the manuscript by performing all additional experiments. We found the comments from the reviewers were very helpful, which guided our revisions, resulting in this much improved manuscript. Because there are limited changes to this manuscript based on the review, changes in text are noted by the **red color texts**.

REVIEWER COMMENTS

Reviewer #1 (Remarks to the Author):

I appreciate the authors to correctly answer all my concerns.

Response: We were delighted to read that the reviewer's concerns have been resolved.

Reviewer #2 (Remarks to the Author):

The authors have substantially revised and thereby improved their manuscript. However, some points were not addressed properly so I would like to spell them out again.

1) While the dose and the route of application of ACE inhibitors have been clarified, it still remains open why this particular dose was chosen. Is this comparable with plasma or tissue concentrations detected in humans treated with ACEi?

Response: We thank the reviewer for this comment. We employed the dose and route of application of ACEI mainly with reference to J S Silvestre *et al. Circ Res.*¹ as below. We also referenced to these studies (Jeffery, T. K *et al. Br J Pharmacol.* and Barutta, F. *et al. Br J Pharmacol.*)^{2,3} and investigated the proper application of ACEI in our preliminary experiment.

“Male (8-week-old) knockout mice lacking the bradykinin B2 receptor gene (B2^{-/-}) and wild-type J129Sv/B16 controls (Jackson Laboratory) underwent surgery to induce unilateral hindlimb ischemia. Animals were anesthetized by isoflurane inhaling. The ligature was performed on the right femoral artery, 0.5 cm proximal to the bifurcation of the saphenous and popliteal arteries, as previously described. Wild-type and B2^{-/-} mice (6 animals per group) were then treated with or without ACE inhibitor (perindopril, Servier, France, 3 mg/kg/d) for 28 days. This study was conducted in accordance with both institutional guidelines and those formulated by the European community for experimental animal use (L358-86/609/EEC).”

2) The authors now provide some more information about the sp1/sp3 EC-KO mice (vessel density in muscle, Evans blue permeability, IL1b, TNFa). This information should be given to all readers and not only the reviewer. Please add it to the suppl. figures.

Response: We thank the reviewer for this comment. As suggested, we have added the information about sp1/sp3 EC-KO mice in Fig. S16.

3) The whole manuscript is poorly written and not sufficient for publication. The introduction is a simple description of all of the involved molecules. No connections, no hypotheses, no rationale.. The different parts of the Results chapter are poorly linked with each other. The Discussion is a chaotic mixture of experimental details, superficial associations with some clinical data, and many more.

Response: We thank the reviewer for this comment. As suggested, we have revised the manuscript including part of the introduction, part of the results, and the whole discussion. The changes in text are noted by the **red color texts**.

Reviewer #3 (Remarks to the Author):

The authors have addressed most of the questions properly. However, there are a few concerns for minor revision.

1. The fluorescent signal in Figure S10 is too weak. Improved images are required.

Response: We thank the reviewer for this comment. As suggested, we have revised the Figure S10.

2. a summary of the points in R 3.4 could be added in the discussion part of the manuscript.

Response: We thank the reviewer for this comment. As suggested, we have added a summary of the separate working machinery of Sp1 and Sp3 in discussion.

- 1 Silvestre, J. S. *et al.* Proangiogenic effect of angiotensin-converting enzyme inhibition is mediated by the bradykinin B(2) receptor pathway. *Circulation research* **89**, 678-683, doi:10.1161/hh2001.097691 (2001).
- 2 Jeffery, T. K. & Wanstall, J. C. Perindopril, an angiotensin converting enzyme inhibitor, in pulmonary hypertensive rats: comparative effects on pulmonary vascular structure and function. *British journal of pharmacology* **128**, 1407-1418, doi:10.1038/sj.bjp.0702923 (1999).
- 3 Barutta, F. *et al.* Reversal of albuminuria by combined AM6545 and perindopril therapy in experimental diabetic nephropathy. *British journal of pharmacology* **175**, 4371-4385, doi:10.1111/bph.14495 (2018).

REVIEWERS' COMMENTS

Reviewer #2 (Remarks to the Author):

The authors have addressed all my remaining concerns in an appropriate manner. The manuscript has been substantially improved.

Reviewer #5 (Remarks to the Author):

Minor points

- 1) figure 2A & 7E & 8G: label for measured parameters in staining images is missing - for each horizontal row of images.
- 2) figure 3C & 4D: label for matrigel explants could be added to the images.

We thank the reviewers for their thorough evaluation of this manuscript and the opportunity to strengthen our findings and conclusions. Thanks to these comments, we are extremely encouraged to revise the manuscript by performing all additional experiments. We found the comments from the reviewers were very helpful, which guided our revisions, resulting in this much improved manuscript.

REVIEWER COMMENTS

Reviewer #2 (Remarks to the Author):

The authors have addressed all my remaining concerns in an appropriate manner. The manuscript has been substantially improved.

Response: We were delighted to read that the reviewer's concerns have been resolved.

Reviewer #5 (Remarks to the Author):

Minor points

1) figure 2A & 7E & 8G: label for measured parameters in staining images is missing - for each horizontal row of images.

Response: We thank the reviewer for this comment. We have added label for measured parameters in figure 2A & 7E & 8G.

2) figure 3C & 4D: label for matrigel explants could be added to the images.

Response: We thank the reviewer for this comment. We have added label for matrigel explants in figure 3C & 4D.